# Neuropilin-1: A Multifaceted Target for Cancer Therapy

**DOI:** 10.3390/curroncol32040203

**Published:** 2025-03-31

**Authors:** Sai Manasa Varanasi, Yash Gulani, Hari Krishnareddy Rachamala, Debabrata Mukhopadhyay, Ramcharan Singh Angom

**Affiliations:** Department of Biochemistry and Molecular Biology, Mayo Clinic, Jacksonville, FL 32224, USA; varanasi.saimanasa@mayo.edu (S.M.V.); yashgulani1@gmail.com (Y.G.); rachamala.hari@mayo.edu (H.K.R.)

**Keywords:** neuropilins, semaphorins, VEGFA, tumor biology, cancer therapy, cancer stem cells, immunotherapy, transcriptomics, OMICS

## Abstract

Neuropilin-1 (NRP1), initially identified as a neuronal guidance protein, has emerged as a multifaceted regulator in cancer biology. Beyond its role in axonal guidance and angiogenesis, NRP1 is increasingly recognized for its significant impact on tumor progression and therapeutic outcomes. This review explores the diverse functions of NRP1 in cancer, encompassing its influence on tumor cell proliferation, migration, invasion, and metastasis. NRP1 interacts with several key signaling pathways, including vascular endothelial growth factor (VEGF), semaphorins, and transforming growth factor-beta (TGF-β), modulating the tumor microenvironment and promoting angiogenesis. Moreover, NRP1 expression correlates with poor prognosis in various malignancies, underscoring its potential as a prognostic biomarker. Therapeutically, targeting NRP1 holds promise as a novel strategy to inhibit tumor growth and enhance the efficacy of regular treatments such as chemotherapy and radiotherapy. Strategies involving NRP1-targeted therapies, including monoclonal antibodies, small molecule inhibitors, and gene silencing techniques, are being actively investigated in preclinical and clinical settings. Despite challenges in specificity and delivery, advances in understanding NRP1 biology offer new avenues for personalized cancer therapy. Although several types of cancer cells can express NRPs, the role of NRPs in tumor pathogenesis is largely unknown. Future investigations are needed to enhance our understanding of the effects and mechanisms of NRPs on the proliferation, apoptosis, and migration of neuronal, endothelial, and cancer cells. The novel frameworks or multi-omics approaches integrate data from multiple databases to better understand cancer’s molecular and clinical features, develop personalized therapies, and help identify biomarkers. This review highlights the pivotal role of NRP1 in cancer pathogenesis and discusses its implications for developing targeted therapeutic approaches to improve patient outcomes, highlighting the role of OMICS in targeting cancer patients for personalized therapy.

## 1. Introduction

In cancer research, the quest for effective therapies often hinges on identifying molecular targets that play pivotal roles in tumor progression and metastasis. One such promising target is Neuropilin-1 (NRP1), a multifunctional transmembrane glycoprotein that has garnered significant attention recently because of its diverse roles in cancer biology. This article explores the intricate relationship between NRP1 and cancer, highlighting its potential as a therapeutic target.

NRP1 was initially discovered as a neuronal guidance protein involved in axonal development [1,2]. However, its expression is not limited to neurons; it is also found in various non-neuronal tissues, including endothelial cells, immune cells, and cancer cells [3,4,5,6]. A representative diagram showing NRP1 expressions across different cell types is shown in Figure 1. In the context of cancer, NRP1 has been implicated in several key processes that contribute to tumor progression. NRP1 is a versatile and multifunctional molecule that has gained significant attention as a potential target for cancer therapy. This receptor plays a crucial role in various cellular processes, including angiogenesis [7,8], cell survival [9], migration [10,11], and invasion [5,11], which are essential for tumor growth and metastasis [5]. NRP1 interacts with multiple ligands, such as vascular endothelial growth factor (VEGF), semaphorins, and transforming growth factor-beta (TGF-β), mediating diverse signaling pathways that promote oncogenesis [6]. The overexpression of NRP1 has been observed in numerous cancer types, correlating with poor prognosis and increased tumor aggressiveness [12]. Targeting NRP1 in cancer therapy involves several strategies, including the use of monoclonal antibodies [2], small molecule inhibitors [13], and peptide-based therapeutics designed to block its interaction with ligands and co-receptors [5,14]. These approaches aim to disrupt the NRP1-mediated signaling pathways, inhibiting tumor growth and metastasis. Recent advances in understanding the structural and functional aspects of NRP1 have provided insights into designing more effective and specific inhibitors [5,15]. A recent study published in the *Journal of Clinical Investigation* identifies NRP1 as a co-receptor for nerve growth factor (NGF) and tropomyosin-related kinase A (TrkA) in pain-signaling pathways [16]. Inhibiting NRP1 disrupted NGF-induced signaling cascades, leading to reduced activation of pathways associated with pain sensation. This indicates that NRP1 plays a critical role in facilitating NGF-mediated pain signaling [16]. In summary, NRP1 is a critical player in cancer biology, and its multifaceted role makes it an attractive target for therapeutic intervention. Continued research and development of NRP1-targeted therapies have the potential to impact the treatment of various cancers significantly.

Neuropilins (NRPs) are a member of non-tyrosine kinase cell surface glycoproteins, and they are expressed in all vertebrates. NRP’s are 120–140 kDa type I transmembrane proteins involved in various physiological and pathological processes [17]. NRP1 is observed to be expressed on plasmacytoid dendritic cells (pDCs) [1,18,19] arterial endothelium [20] and a small subset of T regulatory cells (Tregs) found in lymphoid tissue [21] in humans (Figure 1). Recently, there has been a great deal of interest in NRP1 as a mediator of tumor development and progression since it was observed to be extensively expressed in tumor vasculature, where NRP1 over-expression is accompanied by tumor progression and poor clinical outcome [22]. The two isoforms of NRP are neuropilin-1 (NRP1) and neuropilin-2 (NRP2), encoded by distinct genes on different chromosomes that arise by gene duplication. NRP1 is located on 10p12 and NRP2 in the 2q34 chromosome. Each encompasses 17 exons with a similar domain structure stalked by a transmembrane region and a cytosolic tail. Since their discovery, numerous NRP isoforms have been reported.

The extracellular part of NRP1 has three unique domains, a1/a2, b1/b2, and c (Figure 2). The a1/a2 portion is homologous to noncatalytic regions of complement components C1r/C1s, interaction domains in BMP-1 (bone morphogenetic protein), Tolloid (Drosophila dorsal–ventral patterning protein), and the carbohydrate-binding proteins involved in the binding of the spermatozoa to the zona pellucida. The b1/b2 domain characteristics of coagulation factors and other proteins are known to bind with anionic phospholipids on the cell surface, leading to cell-cell adhesion [23]. This is where ligands such as semaphorins and VEGFs interact with the MAM (meprin, A5, μ-phosphatase) domain for heterodimerization. The b1/b2 domain is homologous to C1/C2 domains of coagulation factors V and VIII, a discoidin 1-like domain in receptor tyrosine kinase (DDR). The c domain is the central portion and has a conserved amino acid sequence designated as the MAM domain contained in the Metallo endopeptidases memprins and receptor protein tyrosine phosphatase [24]. The post-translational NRP modifications are mediated by the covalent addition of glycosaminoglycan (GAG) attached to a single conserved by Ser^612^ residue of the b-c linker region of the receptor [25]. This modification enhances NRP1’s affinity to bind VEGF. HS-GAG binds to multiple NRPs, forming a cluster which, after binding to VEGFR2, forms a more stable complex, resulting in enhanced signaling by preventing internalization of the complex [25,26]. 

NRP1 is a cell surface glycoprotein initially characterized as a neuronal guidance molecule. NRPs predominantly function as co-receptors for Class III semaphorins and for the VEGF family, two architecturally unrelated classes of ligands with different biological functions [27]. These ligands are widely known for their role in various physiological processes, such as cardiovascular, neuronal development and patterning, angiogenesis, lymphangiogenesis, and numerous clinical disorders [28]. Interestingly, other roles for NRPs occur with myeloid and lymphoid cells in normal physiological and separate pathological situations involving cancer, immunological disorders, and bone diseases [26].

NRP1 is expressed by endothelial cells and tumor cells as an isoform-specific receptor for VEGF, and the extracellular domain of NRP1 allows it to bind to VEGF and act as a co-receptor for VEGF, enhancing the binding affinity and bioactivity of VEGF. Thus, NRP1 mediates VEGF extracellular migration [29]. Recent reports also question the role of the VEGF receptor as a coreceptor for NRP1 in endothelial cell migration and survival. NRP1 has been reported to mediate endothelial cell chemotaxis, cell adhesion, axon guidance, and transduction of multiple signaling pathways. The loss of NRP1 has tremendous effects on growing embryos. For example, embryonic deletion of NRP1 leads to severe vascular defects and embryonic lethality. NRP1 deficiency in adult mice exhibited an attenuated response to semaphorin 3A [30], decreased VEGF-induced vascular permeability [31], and reduced inflammatory responses [4,32]. Some studies suggest upon TGFβ stimulation, NRP1 regulates differential SMAD protein signaling. The pro/anti-angiogenic function of NRP1 shows a novel pathway in cancer metastasis mediated by KRAS and TGF-β signaling through NRP1 [33]. In this review, we have emphasized NRP1 and its role in signaling between VEGFA and semaphorins, its role in immune cells, and the tumor microenvironment. We have further discussed the detailed role of NRP1 in various disease conditions. Importantly, we have included an OMICS-based approach to understanding the role of NRP1 in cancer-associated cellular processes and the possibility of NRP1 as an essential target for cancer therapy using online databases such as cBioportal for cancer genomics [34], University of Alabama at Birmingham CANcer data analysis Portal (UALCAN) [35], Enrichr [36], Gene Expression Profiling Interactive Analysis (GEPIA2) [37]. A summary of various ligands binding with NRP1 has been described in Table 1. This table explains the various NRP1 domain interactions with extracellular ligands. For example, we explained that Sema3A binds to the NRP1 CUB extracellular domain and promotes prolonged T-cell and DC interaction and T-cell activation by IL-10 secretion [24,25,26].

## 2. VEGFA and NRP1 Signaling

VEGFA is a member of the growth factor family, including VEGF-B, VEGF-C, VEGF-D, and placental growth factor (PIGF). VEGFA exists in many variants, such as VEGF121, VEGF145, VEGF148, VEGF165, VEGF183, VEGF189, and VEGF206. VEGF binds to multiple tyrosine kinase receptors, such as VEGFR1, VEGFR2, and VEGFR3. VEGFR2 is the main receptor that conveys VEGF signals in endothelial cells. This binding promotes autophosphorylation on several tyrosine residues, which bind to intracellular adaptor proteins to initiate signal transduction [50].

The role of NRP1 in modulating the VEGF-dependent neural signaling pathways has been studied using VEGF164 isoform in many murine studies. The VEGF164 isoform signaling through NRP1 has been studied in many different types of neurons. These VEGF164-NRP1 signaling does not require the VEGFR2 to promote the signals, suggesting the role of NRP1 as the major driver [51]. Further, NRP1, as a co-receptor for VEGFR2, was first identified in porcine endothelial cells. VEGF165-induced signal transduction activation involves ERK1/2 and p38 Map kinase (p38MAPK). These suggest the role of NRP1 in VEGF-dependent vascular signaling pathways [52].

Many studies have been conducted to understand the central role of NRP1 as a regulator for VEGF-dependent arteriogenesis and angiogenesis. Arteriogenesis is a process by which small blood vessels give rise to functional arteries, dependent on VEGF via arterial fate specification, lumen expansion, and endothelial cell proliferation. VEGF165 binds to VEGFR2, which activates downstream signaling of ERK1/2 along with Synectin. NRP1 is required to bridge this VEGFR2 to Synectin. Mechanistically, the NRP1 cytoplasmic tail and Synectin link VEGFR2 to myosin VI-mediated transport machinery and promote VEGFR2-enabled downstream dephosphorylation, enabling the sustained ERK activation for arteriogenesis [53].

NRP1 also plays a crucial role in angiogenesis. Chemotactic VEGF164 signaling through VEGFR2 was studied during angiogenesis. The endothelial cells sprout from pre-existing vasculature, are composed of stalk cells, and are led by high migratory endothelial cells, termed tip cells. Tip cells extend filopodia into the extracellular environment and are highly responsive to VEGF and NRP1. This has been studied in detail in developmental angiogenesis in zebrafish larval trunks [54]. These involve the activation of ERK and p38 pathways downstream. NRP1 also activates ABL1 and CDC42 in response to integrin ligands in the extracellular matrix (ECM) to promote remodeling and filopodia to enable changes in shape. NRP1 also suppresses TGF-β signaling to promote tip cell positioning through an incompletely understood mechanism [55].

The role of VEGF in the therapeutic potential to restore blood flow in many cardiac tissues is mainly balanced by the risk of hyperpermeability that causes tissue injury. VEGF regulates vascular permeability apart from angiogenesis and arteriogenesis. Many studies have been conducted to study the vascular permeability effect of VEGF, and NRP1 contributes to VEGF-induced vascular hyper permeability [56]. Some in vivo studies explain that the genetic deletion of NRP1 in endothelial cells impairs skin permeability after Sema3A intradermal injection, which further raises a possibility about the contribution of NRP1 in VEGF-mediated vascular hyperpermeability. VEGF and Sema3A-independent NRP1 signaling pathways in the vasculature have also been studied in many animal models, demonstrating that NRP1 is essential for endothelial tip cell function and promotes angiogenesis in VEGF-independent pathways [55]. One of the remarkable studies by our group for the first time reported that VEGF exercises an angiogenesis-independent role in cancer cells to help their malignant advancement. This study by Cao et al. 2012 has defined a novel function for VEGF in the dedifferentiation of tumor cells, opening a new avenue in the role of NRP1 in cancer in addition to its known proangiogenic function [57].

## 3. Semaphorin-NRP1 Signaling

Semaphorins are products of a large gene family containing 28 genes, 21 of which are found in vertebrates. Class III semaphorins make up a subfamily of seven vertebrate semaphorins, which are unique in that they are only secreted and contain a basic domain at their C termini. The semaphorin family, containing thirty proteins, is further divided into eight classes based on structural features and distribution. Classes 1, 4, 5, and 6 are transmembrane proteins, Classes 2, 3, and V are secreted proteins, and Class 7 are glycosylphosphatidylinositol (GPI)-linked proteins. Of all semaphorins, Class III semaphorins are more prominently studied in various physiological processes. Among the Class III semaphorins, Sema3A exclusively binds to neuropilin-1. Sema3F and Sema3G bind to neuropilin-2, and Sema3B, Sema3C, and Sema3D bind to both. Neuropilins transduce semaphorin signals by forming complexes with one or more of the four type-A plexins or plexin-D1 [58]. The functional significance of Sema3A signaling through NRP’s has been widely studied in neurons, vasculature, rodent models of neovascular eye pathology, and acute vascular permeability [4]. Sema3B and Sema3F act as tumor suppressors in some cancers by suppressing tumorigenesis in certain adenocarcinoma cell lines. Sema3B also functions as an angiogenesis inhibitor in small lung cell lung cancer cell lines [59]. Sema3B in small lung cell lines showed that the suppression is through the induction of apoptosis with a possible association of angiogenesis inhibition [60]. On the other hand, Sema3F can interact with NRP1 and reduce the growth of metastatic activity in colorectal carcinoma cells by modifying the integrin αvβ3 region [61] Sema3G’s role in cancer and signaling is less known, except that it plays a crucial role in binding to NRP2 and has been identified as a prognostic marker in glioma cells [62]. In order to study the interaction network between NRP1 and NRP2 with the Class III semaphorin family, a STRING analysis has been performed, as shown in Figure 3 by using the STRING database [63]. Similar to the plexin family, Class III semaphorins are distinguished by the presence of 500 amino acid sema domain close to the N-terminus. This domain influences the specificity of receptor binding. The Sema domain has a cleavage site activity, which, when cleaved, inactivates the Class III semaphorins [64]. Semaphorins bind to one or both receptors of the neuropilin family, as these NRPs are short intracellular domains that transduce a signal. Class III semaphorins bind to the plexin family, which function as signal transducing components [65]. NRPs bind to diverse ligands (such as VEGFA, VEGFB, VEGFC, PLGF, and TGFB) and many types of membrane-bound receptors (such as VEGFR1, VEGFR2, and VEGFR3). Class III semaphorins compete with VEGF family for binding with NRPs. A recent study by Swati et al., discusses the importance of NRP1 binding to VEGFR2 and endoglin (ENG), by forming a tripartite complex. VEGFA mediated phosphorylation requires ENG and NRP1 and it provides an attractive therapeutic target for modulation of VEGFA signaling [66]. By altering the binding sites, the post-translational changes in semaphorins, such as the cleavage of furin-like protein convertases, prevent NRPs from binding [67,68]. The plexin family has nine receptors: four type A plexin, three type B plexin, and a single C and D plexin. Four type A plexin and plexin D1 form complexes with NRPs. The plexins represent two domains: the extracellular domain, which serves as the auto-inhibitory domain, and the intracellular domain, characterized by the presence of GTPase activating protein (GAP domain) [69]. Plexins also contain intracellular tyrosine phosphorylation sites. These are mostly in inactive forms. The binding of Sema proteins leads to a huge spatial organization change in plexin dimers in both intracellular and extracellular domains, shifting their conformation from inactive to active form [70,71]. For instance, in Class III semaphorins Sema3A, data indicate that overexpression of plexin A2 and NRP1 results in the formation of tetrameric Sema3A receptors, which are made up of two plexin A2 receptors and two NRP1 receptors after Sema3A activation [38]. A recent study conducted by Mike B. Barnkob et al. shows the role of Sema3A in anti-tumor immunity, they observed that NRP1 and PlexinA1 and Plexin A4 are upregulated in tumor specific CD8+ T cells and NRP1 deficiency enhances anti-tumor activity of CD8+ T cells against Sema3A rich tumors [72].

## 4. NRP1 in Cancer

NRP1 is involved in various pathways that promote tumors by activating diverse genes. Studies on renal carcinoma cells have shown that NRP1 is involved in dedifferentiation. NRP1 binds to Plexin A4 and Sema3A mounting immunoinhibitory signals, enhances epithelial–mesenchymal transition (EMT) followed by cell migration and invasion leading to metastasis [42,73,74]. Angiogenesis has proven to be a prime regulator of tumor growth. Tumor angiogenesis is a control point in metastasis and the expansion of tumor cells to other sites. By characterizing the molecular regulators of angiogenesis, whether inhibitors or stimulators, we can focus on potential targets to slow growth and metastasis. VEGF is among the best-characterized angiogenesis regulatory systems and has proven to be a likely therapeutic target for metastasis prevention. Studying its novel receptor, NRP1 has proven to be a likely therapeutic target for preventing metastasis and slowing tumor growth [75]. Studies reported by our lab reveal a novel crosstalk between NRP1 and TNFα in vascular endothelial cells. ShRNA-mediated NRP1 knockdown suppressed TNF-stimulated leukocyte adhesion and expression of Intracellular adhesion molecule-1 (ICAM-1) and Vascular cell adhesion molecule-1 (VCAM-1). TNFα was shown to downregulate NRP1 through the TNFR1–NFκB pathway in Human umbilical vein endothelial cells (HUVECs) [76]. Another study by our group demonstrated that NRP1 knockdown in cancer cells depends on the genetic status of KRAS. ShRNA-mediated NRP1 knockdown in KRAS wildtype cells increased cell viability and tumor growth by decreasing the SMAD2 phosphorylation, whereas a KRAS mutant reverses the increased viability and leads to tumor inhibition. Our lab has studied this in both pancreatic and lung cancer [77]. Furthermore, our lab showed that the disruption of the VEGF/NRP1 axis activates proangiogenic and protumorigenic signaling in endothelial and tumor cells via Tumor targeting peptides (TTP); liposomes encapsulating both everolimus (a small molecule inhibitor) demonstrated tumor growth retardation in ccRCC models, improving overall therapeutic outcome in ccRCC patients [78].

Contrasting new evidence has shown that NRP1 can also be considered an angiogenesis mediator in tumors, suggesting that tumor growth and metastasis may be the product of molecular interactions between NRP1’s originally deduced process of neuronal guidance and angiogenesis [75]. NRP1 is expressed by Endothelial cells (EC) both in vitro and in vivo, and when expressed alongside VEGFR-2, VEGF165 binding was seen to be enhanced, and F165 mediated EC chemotaxis [75]. In the absence of VEGFR2, EC did not respond to any VEGF isoform, suggesting that NRP1 was not a signaling receptor for chemotaxis and that in EC, NRP1 acted as a co-receptor for VEGFR-2 [75]. Over-expression of NRP1 in transgenic mice resulted in lethal embryonic abnormalities, including incorrect nerve fiber sprouting and cardiovascular and skeletal system abnormalities because of excess capillary and blood vessel formation [75]. Targeted disruption of the NRP1 gene also resulted in embryonic lethality, attributed to defects in the cardiovascular system such as impairment of neural angiogenesis, lack of branchial arch-related great vessels formation, dorsal aorta transposition of great vessels, and insufficient development of vascular networks in the yolk sac. This vascular regression seen in NRP1 deficient embryos compared with the overproduction of vessels in NRP1 overexpression mice suggested that NRP1 plays a role in not only nerve fiber guidance but also embryonic blood vessel formation [75]. Tumor cell lines such as prostate and breast carcinoma cell lines express abundant NRP1, nearly 1–2 × 10^5^ receptors per cell [75]. NRP1 is now known to be a tumor promoter, and several studies have been conducted, but effective targeting of NRP1 as a potential therapeutic target has not been achieved to date [79]. Various trails using monoclonal antibodies targeting NRP1 inhibitors and clinical trials in renal carcinoma patients using bevacizumab and sunitinib combinations resulted in numerous side effects for patients [80,81]

## 5. NRP1 Promotes Angiogenesis, Tumor Proliferation, and Migration

Angiogenesis and vascular maturity play a prominent role in tumorigenesis and tumor development. NRP1 plays an important role in angiogenesis, cell survival, migration, and invasion, making it an eligible marker for tumors. Class III semaphorins such as Sema3A, Sema3B, Sema3E, Sema3F, and Sema3G are studied in angiogenesis [82]. Sema families are linked to modulation in several cancer processes, such as angiogenesis, cancer cell invasiveness and metastasis, and cancer cell survival [5,6]. Sema3F and Sema3B act as tumor suppressors, whereas Sema3F and Sema3B are involved in cancer progression [83]. Sema3A/NRP1 signaling, when blocked, can inhibit angiogenesis [84]. NRP1/Sema4A can regulate the stability of T cells in certain inflammatory sites [41]. However, underlying molecular mechanisms remain unclear. A recent study by Peng Gao and colleagues explained that Signal transducer and activator of transcription 6 (STAT6) promoted angiogenesis by increasing the expression of NRP1 in endothelial cells. STAT6 inhibitor and STAT6 siRNA reduced EC proliferation, migration, and tube formation by drastically downregulating NRP1. The luciferase assay and chromatin immunoprecipitation demonstrated that STAT6 could bind to human NRP1 and increase promoter activity, explaining the role of STAT6 as a potential target for anti-angiogenesis therapy [85]. NRP1 activation through cross-talk between luminal progenitor cells, endothelial tip cells, and immature perivascular cells via the Pleiotrophin–NRP1 axis in the pathogenesis of Inflammatory Breast Cancer (IBC) was recently studied, which correlated with increased angiogenesis and IBC metastasis [86]. Biophysical interaction studies between endoglin (ENG), NRP1, and VEGFR2 were conducted to understand the VEGFA-induced signaling in endothelial cells prior to ligand stimulation [66]. Gain and loss of function experiments determined the role of NRP1 in tumor progression using The Cancer Genome Atlas (TCGA) database in vascular endothelial cells. RNA Seq analysis was performed to gain an understanding of the proangiogenic pathways. The mRNA levels of NRP1 significantly increased in primary tumors compared with normal cells, highlighting the functioning of NRP1 in EC proliferation, motility, and capillary tube-like formation and in reducing apoptosis. Overexpression of NRP1 decreased these EC markers' expression levels (PECAM-1, MMP-9, and PIGF) and drastically reduced vascular maturity. Sema4D was revealed to be a key protein associated with NRP1 in this study [87]. The primary causes of the bad prognosis are tumor infiltration and migration. When VEGFA binds to NRP1, it encourages Ras homolog family member A (RhoA) activation, which subsequently contributes to the degradation of cyclin-dependent kinase inhibitor (p27kip1), which in turn encourages the proliferation of tumor cells. This has been proven in glioblastoma (GBM), prostate cancer, and skin cancer [88,89]. Recent studies on Bladder urothelial carcinoma (BC) investigated the functions and mechanisms of NRP1 in BC pathogenesis and progression. NRP1 expression was high in BC tissues, and knockdown of NRP1 promoted apoptosis, suppressed proliferation, angiogenesis, migration, and invasion of BC cells. Differentially expressed genes (DEGs) induced by NRP1 silencing, followed by GO/KEGG, IPA bioinformatics analysis revealed MAPK signaling and other pathways such as antiviral response, immune responses, cell cycle proliferation, and migration of cells are highly affected, indicating the role of NRP1 in tumor progression and migration [90]. The activation of NRP1 can be accomplished via Sema families involved in the progression of tumors. Sema3C has recently been studied as a novel biomarker in Hepatocellular carcinoma (HCC), where NRP1 and Integrin Beta1 (ITGβ1) act as functional receptors of Sema3C which activates Serine threonine Kinase (AKT)/GLI Family Zinc Finger 1 (Gli1)/c-Myc signaling pathways leading to tumor initiation [91]. Another study conducted by Lijuan Yin and his colleagues explains the role of monoamine oxidase (MAOA) is a functional mediator of perineural invasion in prostate cancers. MAOA activates Sema3C during transcription and stimulation of cMET, and this activation is co-activated by PlexinA2 and NRP1 [92]. Transcriptome profiling in prostate cancers revealed NRP1 correlated with unfavorable prognosis. The b1/b2 domain of NRP1 interacts with the extracellular domain of Epidermal growth factor receptor (EGFR), activating EGFR via the AKT pathway and promoting the proliferation and migration of PC cells. Hypoxia Inducible Factor 1α (HIF1α) is involved as a transcriptional regulator of NRP1 in this study [93]. lncRNA’s can regulate gene expression both at transcription and post-translational levels, and many of these have played a prominent role in cancer progression [94]. In vivo experiments on zebrafish xenografts pertaining to the progression of colorectal cancer (CRC) through the miR-24-3p/NRP1 axis have predicted poor prognosis. Long non-coding RNA plasmacytoma variant translocation (PVT1) binds to NRP1 and regulates its expression but knock down of PVT1 suppresses the growth and metastasis of Circulating Endothelial cells (CEC), explaining the role of NRP1 [95]. In addition to directly encouraging tumor development and migration, NRP1 also modifies the tumor microenvironment by interacting with integrins and altering the extracellular matrix to affect tumor growth [96]. Microarray and proteomics studies on Osteosarcoma (OS) revealed NRP1 as a major cancer cell migration and invasion regulator. In vivo experiments in lung metastatic models for knockout NRP1 significantly mitigated lung metastasis. This has downregulated the migration machinery such as S locus receptor kinase (SRC), Focal Adhesion Kinase (FAK), and Rho-associated coiled-coil containing protein kinase 1 (ROCK1) expression relating NRP1 interaction with ITGβ1 [97].

## 6. Other Regulatory Role of NRP1

Recent studies about the NRP1 in apoptosis are reported in Bladder urothelial carcinoma. NRP1 knockdown promoted apoptosis and suppressed proliferation, angiogenesis, migration, and invasion, indicating NRP1 as a tumor promoter. Using GO/KEGG and IPA analysis, it was found that cell cycle arrest is a primary mechanism for inducing apoptosis. Protein expression levels for BIRC3 and CDK6 were upregulated after NRP1 knockdown, and CDK4, CCNE2, FOS, and CDK2 were downregulated [90]. NRP1 downregulation progressed BC, which is associated with the activation of MAPK signaling and activation of another molecular mechanism [90]. Another study by Nadejda and colleagues reported the overexpression of VEGF isoform (i.e., VEGF189) in MDA-MB-231 breast cancer cells induces apoptosis through NRP1. Overexpression of this VEGF189 isoform increased apoptosis, whereas knockdown of NRP1 in VEGF189 reduced apoptosis and increased necrosis in these cell lines under stress conditions [98]. Another form of immunogenic cell death in cancers has been studied called necroptosis, where they generate second messengers such as damage-associated molecular patterns (DAMPs) that act on immune cells, triggering inflammation and thereby attracting immune cells in the tumor microenvironment. Necroptosis activates antigen-presenting cells and drives cross-priming of CD8+ T cells, inducing antitumor immune responses [99]. Necroptosis activates serine/threonine kinases, which in turn phosphorylate kinase domains such as pseudokinases, triggering conformational changes and facilitating oligomerization, translocation, and disruption of cellular membranes, leading to lytic cell death [99]. Other types of immunologic cell death, such as pyroptosis, oxytosis, mitochondrial permeability-driven necrosis, and their mechanisms in cancers, should be studied in detail. Other forms of cell death, such as ferroptosis involving iron and mitochondria-dependent ferroptosis and copper and mitochondrial respiration-dependent cuproptosis, are studied in cancers apart from programmed cell death. Recent research on gastric cancer describes the role of cuproptosis and immune cell regulation in TME. From KEGG analysis and differentially expressed genes, NRP1, CXCR4, LGR6, and CTLA4 are said to be the key regulators of cuproptosis. Cuproptosis-related immune genes (CRIGs) are now considered to predict patient prognosis by understanding the intrinsic and extrinsic mechanisms between these genes [100]. Another research finding in ccRCC patients has shown a correlation between ferroptosis-related long non-coding RNA (lncRNAs) and the prognosis of patients. Immune checkpoint variability analysis has identified T-cell immunoglobulin and mucin domain containing-3(HAVCR2), NRP1, Human Endogenous Retrovirus H Long terminal repeat associating protein 2 (HHLA2) genes to be upregulated in low-risk group, while CD44, Glucocorticoid-induced tumor necrosis factor receptor (TNFRSF18), CD276, TNFRSF25 were upregulated in high-risk group providing new direction for clinical treatment [100]. Ferroptosis and Cuproptosis are relatively new areas that need to be extensively explored for their role in cancer progression or suppression. Further studies aimed at identifying the important biomarker genes for early detection of cancers in these areas could be of potential interest to our lab.

## 7. Role of NRP1 in Immune Cells

The immune system comprises two compartments: the innate and adaptive systems. The innate immune system mainly comprises myeloid lineage cells, macrophages, DC, neutrophils, eosinophils, basophils, and natural killer (NK) cells, whereas the adaptive arm includes T and B cells. A complex interplay between the immune cells is crucial for controlling infectious diseases and neoplasia. One key mechanism through which NRP1 influences immune cells is its role in immune cell migration. NRP1 has been shown to modulate the chemotaxis of immune cells in response to chemokines and other guidance cues. For example, NRP1 can enhance the migration of dendritic cells toward lymph nodes, facilitating antigen presentation and immune priming [101]. Additionally, NRP1 can promote the homing of regulatory T cells to inflammatory sites, potentially dampening excessive immune responses [42,102]. NRP1 can additionally modulate immune cell activation and function. NRP1 engagement can impact T cell receptor (TCR) signaling and cytokine production in T cells. It has been suggested that NRP1 can act as a co-receptor for TCR signaling, influencing T cell activation thresholds. NRP1 has been implicated in stabilizing the suppressive function of regulatory T cells, contributing to immune tolerance and homeostasis. Additionally, NRP1 can participate in immune checkpoint regulation. The interaction between NRP1 and its ligands, such as Sema3A, can influence the expression and function of immune checkpoint molecules such as programmed cell death protein 1 (PD-1) on T cells, affecting immune responses in cancer and autoimmunity. Exploring NRPs in recent years has revealed that they are expressed in diverse subsets of immune cells and are focal for regulating immune response. NRP1 and NRP2 are shown to express on DCs, macrophages, T cell subpopulations, and mast cells where they regulate immune responses [26], as listed in these have been summarized in Table 2. NRP1 expression has also been reported in immature thymocytes [103]. DCs expressed NRP1 can be trogocytosed to T cells, where the T cells start expressing NRP1 within 15 min of coculture with DCs. NRP1 is also specifically expressed on a subset of T follicular helper (Tfh) cells in human secondary lymphoid organs and is linked with B cell differentiation [104]. Chang Liu reported that CD8+ T cell-restricted NRP1 deletion exhibited substantially enhanced protection from tumor rechallenge and sensitivity to anti-PD1 immunotherapy, making NRP1 a unique immune memory checkpoint [105]. NRP1 is also considered a marker for murine T_reg_s, where its expression is correlated with immunosuppression [102]. Conversely, NRP1 expression as a marker for human T_reg_s is still under dispute and is purported to be able to classify between thymic-derived and mucosa-generated peripherally derived T_reg_ cells [106]. Sema3A binds to Plexin A requires NRP1, found on CD4+ regulatory T-cells and tumor-infiltrating CD8+ T cells [107]. Sema3A is produced in cancer cells and dysfunctional tumor-specific CD8+ T cells and can modulate anti-tumor response by upregulating NRP1, such as PD-1 [105,107,108]. Mike B Barnkob and his team reported that NRP1 KO mice and genetic models of Sema3A in cancer cells, T cell expression of NRP1, and tumor cell expression of Sema3A can control CD8+ T cell infiltration. In vivo experiments also prove that Sema3A can strongly affect CD8+ T cell movement and migration, forming a synapse [72]. On the other hand, Sema3G, one of the quietest members of Sema family, has recently been studied as key regulator of immune responses in cancer. Sema3G has a significant higher binding affinity to NRP1 than NRP2 and blocking Sema3G using CRISPR/Cas9 technology has restored cytotoxicity of CD8+ T cells and inhibited growth of tumors in vivo [109]. The loss of NRP1 because of Treg cells fragility with the expression of Interferon Gamma (IFNδ) has been studied to retain immunosuppressive activity and increase efficacy in immunotherapies [110]. Treg cells help tumor cells to escape killing cytotoxic CD8+ T cells causing an immunosuppressive effect; in patients with PAAD, tumor microenvironment (TME) and poor prognosis have correlated with high expression of NRP1, indicating more CD8+ T cells [111]. Using GO/KEGG analyses, GSEA, and in vitro experiments, NRPs exert protumor effects via immune-related such as the PI3K-AKT and EMT signaling pathways and independent interactions such as hematopoietic cell lineage regulation, myeloid leukocyte migration and myeloid immune cell infiltration in PAAD patients [111]. Table 2 shows various functions of the NRPs in the immune cells, explaining the role of dendritic cells, macrophages, and T cells in antigen processing and presentation.

## 8. NRP1 in Various Cancers

### 8.1. Glioblastoma (GBM) and NRP1

GBM is a highly proliferative and locally invasive cancer with an elevated decline rate and poor diagnosis [33]. GBM exploits the neural microenvironment via angiogenesis for prolonged survival. Anti-VEGF antibodies such as VEGFA blocking antibody bevacizumab have proven to provide short-term solutions because of blood vessel regression and stabilization of vascular permeability, but tumor recurrence is a very common event [119]. NRP1 is expressed in GBM cells and promotes TGFβ receptor internalization and signaling via Smad transcription factors. In recurrent GBM, NRP1 is down-regulated, indicating that disruption of the balance between VEGF-A and TGFβ signaling is a mechanism that promotes anti-angiogenic resistance in GBM [119]. The TGFβ pathway is activated during the progression from low-grade astrocytoma to GBM by tumor cells to promote proliferation. The deposition of latent-TGFβ ligands in the extracellular matrix and their activation by avβ8 integrins and other pathways stimulates TGFβ signaling in GBM cells. Also, microglia express TGFβR2 in response to neuroinflammation, further promoting angiogenesis pathologies. Our group has shown that NRP1 ablation of patient-derived GBM cells enhances the overall survival in tumor-bearing mice [120]. Patient-derived GBM cells express shRNA of VEGF or NRP1 attenuate cancer stem cells (CSCs), inhibiting the tumor-initiating cell’s neurosphere forming capacity and migration. Our lab previously established results that also explain the role of VEGF and NRP1 knockdown inhibiting patient-derived GBM xenografts in both zebrafish and mouse models, whereas the later showed prolonged survival in mice models. Therefore, NRP1 targeting might provide insight into the inhibition of GBM progression [120]. We performed survival analysis using GEPIA2; the survival graphs were directly extracted from this database. Figure 4 shows the disease-free survival and overall survival of GBM patients with respect to NRP1. The high NRP1 expression groups show lower survival compared with low NRP1 expression groups (*n* = 81).

### 8.2. Pancreatic Cancer and NRP1

Pancreatic ductal adenocarcinoma (PDAC) continues to be one of the move devastating diseases with a poor prognosis and rising incidence [121]. PDAC leads to more than 90% of pancreatic malignancies worldwide and is considered the fourth most common cause of cancer-related deaths, with a 5-year survival of less than 8%. The overall survival in patients with PAAD (pancreatic adenocarcinoma) was extracted from GEPIA2, which shows no difference between low and high NRP1 groups, as presented in Figure 5. PDAC tumor stroma/desmoplasias house many cancers-associated fibroblasts (CAFs) and other stromal cells that can alter the tumor microenvironment by releasing oncogenic and angiogenic factors [23]. In a normal pancreas, there is an absence of NRP1 and a presence of NRP2 in the endocrine islets and acinar cells, while in pancreatic cancer cells, both NRP1 and NRP2 are highly expressed. Increased NRP1 expression is associated with tumor angiogenesis, advanced tumor–node-metastasis, pT stage, node invasion, and dismal postoperative survival. It has been discovered that Mucin 1, a transmembrane glycoprotein, is associated with NRP1 and responsible for inducing VEGF signaling and angiogenesis in the tumor environment [122].

### 8.3. Renal Cancer and NRP1

Renal cell carcinoma constitutes nearly 3% of all adult cancers worldwide [123]. There are as many as 337,860 new cases per year and approximately 150,000 related deaths annually. Approximately 25% of patients with RCC internationally have advanced disease (locally invasive or metastatic disease) diagnosis, 30% of whom cancer recurs after resection [123]. Treatment of Renal Cell Carcinoma (RCC) has proven to be highly challenging once metastasis has ensued, resulting in a 5-year survival of around 13% of patients [123]. The overall survival of KIRC (Kidney Renal Cell Carcinoma) patients with low and high NRP1 shows that the low NRP1 group has a higher survival percentage than the high NRP1. This image is extracted from GEPIA2 [37]. Figure 6 shows the disease-free survival and overall survival of patients with Renal cancer with respect to NRP1.

The two main types of RCC are clear cell and papillary. Clear cell (ccRCC) constitutes nearly 70% of all cases and is characterized by inactivating mutations in the von Hippell–Lindau gene (VHL), which encodes the VHL protein, a critical component in the targeting of the transcription factor hypoxia-inducible factor (HIF) for degradation. HIF remains stable in tumors with VHL mutations and promotes the expression of genes such as VEGF. NRP1, expressed by neuronal, epithelial, inflammatory, and tumor cells, binds VEGF to form a ternary complex with VEGF on endothelial cells, stimulating downstream signaling, which induces proliferation and migration [73]. Research in our lab indicates that the rapamycin (mTOR) pathway plays a crucial role in VEGF synthesis, and its disruption along the VEGF/NRP1 axis has been linked to the activation of proangiogenic and protumerogenic signaling in both endothelial cells and ccRCC tumor cells [124]. Studies from our lab show that the target of VEGF synthesis and the disruption of the VEGF/NRP1 axis is known to activate proangiogenic and protumorigenic signaling in endothelial cells and ccRCC tumor cells. Using TTP (tumor-targeting peptides) liposomes encapsulating Everolimus and small-molecule inhibitor (EG00229) demonstrated high in vitro and in vivo growth retardation that single drug-loaded liposomes leading to a noticeable reduction in lung metastasis in vivo. Therefore, targeting the VEGF signaling in ccRCC via a bifurcated approach using TTPs is beneficial as it has an anti-proliferative effect [78].

### 8.4. Melanoma and NRP1

NRP1 is expressed in tumor-associated vessels and various cancers, indicating a role in tumor progression. A recent study shows that NRP1 expression has been identified in the blood vessels in more than 98% of cancer cases. In contrast, its expression in cancer varies depending on the tissue origin, histological sub-type, and stage [125]. NRP1 overexpression correlates with tumor aggressiveness, advanced disease stage, and poor prognosis [2,125]. The overall survival of patients in SKCM (Skin Cutaneous Melanoma) with high and low NRP1 groups shows no difference. Figure 7 shows the overall survival and disease-free survival analysis for KIRC patients with respect to NRP1. Increased level of NRP1 is associated with tumor invasiveness and metastatic potential [126], for instance, in melanoma and breast cancer [127,128]. NRP1 is also implicated in mediating the effects of VEGF-A and semaphorins on cancer cells’ proliferation, survival, and migration [128]. The expression of NRP1 has also been reported by several stromal cells, including fibroblasts and endothelial and immune cells, which can be stimulated by non-VEGF family growth factors and influence tumor progression [129,130]. Many human metastatic lesions derived from melanoma cell lines secrete VEGFA and express its receptors, including NRP1 [131]. NRP1 enhances the activation of a VEGFA/VEGFR2 autocrine loop, which leads to melanoma cell invasion into the ECM [132] through the up-regulation of VEGF-A and metalloproteinases secretion [133]. Moreover, NRP1 over-expression provides human melanoma cells with an increased in vivo growth rate [134].

### 8.5. Breast Cancer and NRP1

The role of NRP1 has been extensively studied in breast cancer. High NRP1 expression is linked with shorter relapse- and metastasis-free survival, explicitly in Estrogen Receptor (ER)-negative breast cancer cohorts. NRP1 inhibition suppressed the expression of cell markers Zinc finger E-box binding Homeobox1 (ZEB1) and Integrin subunit alpha 6 (ITGα6). NRP1 was required to maintain maximal RAS/MAPK signaling via EGFR and PDGFR, a hallmark of claudin-low tumors [135]. NRP1 expression might be crucial for activating RAS through receptor tyrosine kinases (RTKs) [136]. High VEGF and NRP1 expression were detected in the cytoplasm of cancer cells. As VEGF and NRP1 expression in breast cancer were associated positively with lymph node metastasis, to understand the role of the VEGF/NRP1 axis in breast cancer, a stable VEGF-silencing Metastatic mammary adenocarcinoma cell lines (MDA-MB-231/shVEGF) cells and NRP1 over-expressing MCF-7/NRP1 cells were generated [137]. Knockdown of VEGF or NRP1 significantly inhibited the proliferation of MDA-MB-231 cells, while NRP1 over-expression significantly enhanced the proliferation of MCF-7 cells. Upregulated of E-cadherin and Occludin, but decreased levels of N-cadherin, Vimentin, and Snail expression were detected in the breast cancer cell lines upon VEGF or NRP1 silencing in the breast cancer cells. The VEGF/NRP1 axis promotes the migration and invasion of cancer cells by enhancing the EMT process. This, in turn, elevates the NF-κB and β-catenin signaling. Further investigations are needed to study the specific effect of targeting NRP1 on the growth of breast cancer [137]. The overall survival and disease-free survival of breast cancer patients with respect to NRP1 is presented in Figure 8.

## 9. Targeting Neuropilin-1 in Cancer Therapy

Given its multifaceted roles in cancer progression, NRP1 has surfaced as a potential target for therapeutic intervention. Several approaches are being explored to disrupt NRP1 function and impede tumor growth: Antibody-Based Therapies [5,80]: Monoclonal antibodies targeting NRP1 have been developed to block its interaction with VEGF and other ligands involved in angiogenesis. These antibodies can inhibit tumor vascularization and potentially enhance the efficacy of traditional anti-angiogenic therapies. Small Molecule Inhibitors: Researchers are also investigating small molecule inhibitors that specifically bind to NRP1 and interfere with its signaling pathways [78,138]. These inhibitors can potentially suppress cancer cell migration and invasion, limiting metastatic spread. Combination Therapies: Combinatorial approaches that target NRP1 alongside conventional chemotherapy, radiation therapy, or immune checkpoint inhibitors are being explored to maximize treatment efficacy. These combination therapies aim to overcome resistance mechanisms and improve patient outcomes by targeting multiple pathways simultaneously.

### 9.1. Anti-NRP1 Monoclonal Antibodies (mAbs)

NRP1 function-blocking mAbs have been developed for its three extra-cellular domains, allowing for specific blockade of its co-receptor functions [139]; since NRP1 is extensively expressed on tumor vasculature, where over-expression promotes tumor progression and angiogenesis, NRP1 has proven particularly valuable as a target for anti-angiogenic therapies [80]. Anti-NRP1 mAbs specific to the b1/b2 domain have been developed, which interact with VEGF165; these mAbs can inhibit VEGF165 isoform-induced tumor growth-promoting angiogenesis [13,140]. An anti-NRP1 mAb, MNRP1685A, is currently in Phase I clinical trials for patients with progressive solid tumors and has been shown to reduce tumor burden significantly via blockade of the VEGF pathway. It proved especially effective when combined with the anti-VEGF mAb, bevacizumab [140]. Other work has identified specific peptides able to block NRP1 interactions with VEGF165 and induce apoptosis of NRP1-expressing tumor cells [141]. The blockade of NRP1 might also exert therapeutic effects by modulating the function of NRP1-interacting growth factors such as PDGF, FGF, EGF, and HGF, which have also been implicated in tumor progression. NRP1 silencing has been shown to impair the activity of these growth factors and inhibit tumor growth in some instances [120], although further investigations are required to verify this.

### 9.2. Cell-Penetrating Peptides (CPP)

CPPs are a promising breakthrough for NRP1 in therapy. Recent reviews described CPPs and their function in cutting-edge cancer and immunotherapies [142]. CPPs produce a C-terminal consensus R/KXXR/K sequence that interacts with NRP1’s b1/b2 domain and causes the CPP to be internalized into cells that express NRP1 through an endocytic “bulk transport” process. The “C-end Rule”, or CendR, states that the consensus sequence must be expressed at the C-terminus end of the CPP. CPPs precisely target NRP1-expressing tissue by using NRP1 as an entrance mechanism into NRP1-expressing cells [143]. Therapeutic drugs and CPPs can be coupled for enhanced delivery and targeted tissue penetration. NRP1 is a protein that is abundantly expressed in a variety of cancers, as was previously mentioned, making CPPs especially useful for anti-cancer therapy since medication success depends on effective tumor infiltration and the minimization of hazardous side effects and needs selective targeting of tumor tissues. Other therapeutic approaches, such as ‘natural regulators’, and secreted sNRP1s, may also help regulate NRP1 proangiogenic activity. The ability of sNRP1s to sequester VEGF165 prevents it from interacting with cancer cells and other cells that express NRP1 [144,145,146]. In murine studies where it hinders the development of tumor vasculature, which is crucial for tumor progression, and in a human study of non-small cell lung cancer (NSCLC), where sNRP1 impaired cancer cell invasiveness in vitro [147,148,149], sNRP1s have been observed to reduce tumor growth. Interestingly, treatment of an anti-NRP1 mAb specific to the b1/b2 domain increased circulating NRP1 in human serum [150]. This circulating NRP1 is believed to be released by membrane “shedding”, and it may sequester VEGF, increasing the effectiveness of anti-NRP1 mAbs. In both a human study of NSCLC and a murine model of hepatocellular carcinoma, small interfering RNA (siRNA)-mediated knockdown of NRP1 reduced the invasiveness of cancer cells in vitro [149,150]. These studies demonstrated that NRP1 can inhibit tumor growth.

## 10. NRP1 as a Biomarker in Cancer Immunotherapy

Cancer treatment in the last decade has hugely been the introduction of T cell-targeted immunomodulators blocking the immune checkpoints Cytotoxic T-Lymphocyte-Associated Protein 4 (CTLA-4), programmed cell death protein 1 (PD1) and PDL1 [151]. Tumor mutational burden (TMB), PDL-1 expression, microbiome, hypoxia, IFN-γ, Extracellular matrix, and molecular and cellular characterization within the tumor microenvironment are related to immunotherapy outcomes [152]. Analyzing the gene expression and spatial organization of the complex and heterogenous tumors and their microenvironment could help find prognostic biomarkers of response to immunotherapy [152]. Radiotherapy, chemotherapy, and immunotherapy have succeeded considerably in various cancer models. PD-1 and CTLA-4 are expressed in the T-cells and negatively regulate tumor immune response. However, the tumor cells exploit these molecules and induce tumor tolerance and T-cell exhaustion. Most patients with advanced cancers do not respond well to traditional treatments and have poor survival. GBM and KIRC are a few examples; also, PD-1 inhibitors (Nivolumab, Pembrolizumab, and Cemiplimab), PDL-1 inhibitors (Atezolizumab, Durvalumab, and Avelumab), CTLA-4 inhibitors (Ipilimumab) are approved by the USFDA for the treatment of cancers. The use of ICIs in 10–30% of patients shows a durable clinical response, highlighting the growing need for identifying biomarkers. Some cancers use combination therapy (e.g., Renal Carcinoma) of ICIs and tyrosine kinase inhibitors, which was found to be more beneficial than monotherapy [153].

Optimal strategies will stimulate CD8+ T cells, concurrently modifying the immunosuppressive cells in the tumor microenvironment (TME), especially the regulatory T-cells (Treg) cells [14]. Treg cells have a critical role in the immune system by blocking autoimmunity, limiting immunopathology, and maintaining immune homeostasis. It is the major barrier to effective anti-tumor immunity [41]. Several studies have focused on the role of Treg cells and the pathways involved in overexpressed and underexpressed Treg function. Neuropilins (NRP1) receptors on Treg cells have gained recognition. Using mice models with the Treg-cell restricted deletion of NRP1, restrained Sema4a ligation of NRP1 and Akt phosphorylation cellularly and at the synapse by phosphate and tensin homolog (PTEN), which increased nuclear localization of the transcription factor Foxo3a. The NRP1-induced transcriptome promotes Treg cell stability by enhancing quiescence and survival factors while inhibiting by promoting differentiation, making the NRP1-Sema4a axis crucial for the intratumoral Treg cell functioning [41]. This, in turn, downregulates the activation of Helios, Interleukin 10 (IL-10), Inducible T cell Co-stimulator (ICOS), and CD73 markers along with C-X-C chemokine receptor 3 (CXCR3), interferon Regulatory Factor 4 (IRF-4), and RORγt (T helper lineage markers).

The loss of NRP1 on Treg cells in mouse models restores antitumor immunity with peripheral tolerance. NRP1+ Treg cells are present across many cancers, both intratumorally and peripheral sites, indicating the capability of intratumoral Treg cell fitness. Recent studies on NRP1 show restricted CD8+ T cell reinvigoration in response to ICIs by providing a barrier in CD8+ T cells mediated tumor immunosurveillance [14]. Treg cell expression of NRP1 differs significantly between mice and humans; NRP1+ Treg cell expression is higher in cancer patients [154]. NRP1+ Treg cells are more functionally suppressive, and the expression of Forkhead Box Protein 3 (FOXP3) and Glucocorticoid-induced TNFR-related protein (GITR) is increased. NRP1 antagonism at the VEGF-binding domain reduced human intratumoral Treg cell suppression by 20% by inducing NRP1 protein internalization [155]. The frequency of NRP1+ Treg cells negatively correlates with disease-free survival, and the number of these cells decreases following therapeutic intervention in both pharmacologic and surgical procedures, which is worth noting [6]

Chang Liu and his team studied the role of NRP1 in CD8+ T cells and suggested that it is an immune checkpoint of T cell memory and targeting NRP1 can facilitate the restoration of durable anti-tumor immunity. It has been reported that the NRP1 expression increases within the CD8+ T cell compartment of the Peripheral blood-derived CD8+ T cells from patients with head and neck squamous cell carcinoma and advanced skin malignancies, and it was correlated with the reduced size of the memory CD8+ T cell pool, poor disease outcome and decrease responsiveness in Immune check blockade (ICB) [105].

The role of tumor-associated macrophages (TAMs) is extensively studied in TME. TAMs act as tumor promoters as well as immune suppressors because they can promote tumor initiation and act as central drivers of the immunosuppressive TME through their expression of cell surface receptors, secreted cytokines, chemokines, and enzymes that regulate the recruitment and function in the Treg crosstalk. TAMs are associated with the process of tumorigenesis, growth, infiltration, and spread and have been associated with tumor angiogenesis and lymphangiogenesis [156]. Abrogation of NRP1 in macrophages inhibits the entry of TAMs into tumors in a hypoxic environment. More studies are suggested to understand the role of NRP1 in TAMs in the adaptive immune response for their anti-tumorigenic roles [84].

## 11. Role of NRP1 in Cancer Stem Cells

Cancer stem cells (CSC) can initiate a new primary tumor or metastasis [118]. CSCs are considered progenitors of differentiated cancer cells, with the capacity for self-renewal, differentiating, and expanding the pool of CSCs [157]. These cells can self-renew and induce heterogeneous aspects of tumors, rendering them resistant to chemotherapy and radiotherapy and lowering patient survival rates and multiple resistance mechanisms [158]. These include drug efflux through ABC transporters, over-activation of the DNA damage response, prosurvival pathways, cell cycle, and cell metabolic alterations [159]. Epithelial–mesenchymal transition (EMT) increases the invasion.

NRP signaling promotes CSC survival and the aggressiveness of tumors. The VEGF-NRP pathways have been studied extensively to understand their functional role in TMEs [157]. VEGF-A is required to maintain the epidermal cancer stem cells (ECS) phenotype by the involvement of NRP1 but not the VEGF receptors. Therefore, inhibiting NRP1 in ECSs can decrease the development and motility of tumor cells [160]. This was studied in detail in breast cancer cell lines. Breast cancer CSCs have high expression of VEGFA and NRP1. Triple-negative breast cancer cell line MDA-MB-231 and the hormone-sensitive breast cancer (MCF-7) cell lines were studied for the level of stemness. A similar correlation was observed in many pediatric brain tumor cases, including glioma stem cells and medulloblastoma stem cells [161]. Similarly, VEGF/NRP2 signaling stimulates stemness in breast cancer stem cells by activating Yes Associated Protein (YAP/TAZ) signaling [161]. This signaling mediates homologous recombination by stimulating Rad51 expression, leading to resistance to chemotherapy and chemotherapy resistance. NRP2 maintains the tumor-initiating cells by stimulating α6β1 integrin; this interaction leads to the FAK/Ras pathway, leading to the activation of GL11. GL11 activation involves the recruitment of BMI-1, a key stem cell factor that favors the increased expression of NRP2 [162]. This activation is mainly seen in breast CSCs.

In addition to the interaction between NRP1 and VEGF, NRP1 and GIPC1 are also involved in the survival of CSCs—interactions between them. In ECS, the NRP1/GIPC1/α6/β4 complex activates the FAK/Src signaling pathway that will allow the stabilization of YAP1/∆Np63α, which leads to ECS survival, invasion, and angiogenesis [163]. NRP1/GIPC1 complex gives a key signal in the deformation character of ECS when forming a complex with Syx, a guanine-RhoA exchange factor in angiogenesis [163]. This formation leads to an increase in the p38 MAPK pathway, as Rho A and MAPK are interdependent, so when NRP1/GIPC1 are suppressed, there is a suppression observed in Rho A and MAPK, leading to loss of the ECS phenotype and reduction in the tumor growth [163].

Further, an increased NRP1 expression has been found to increase the vessel number and cause poor prognosis, while increased expression of NRP2 stimulates tumor progression, and down-regulation of NRP2 expression inhibits colon cancer tumorigenesis [164]. In ccRCC, NRP1 down-regulation decreases migration, invasion, and tumorigenesis, whereas NRP2 downregulation decreases cell extravasation [165]. Interestingly, the downregulation of NRP1 reduces cell migration, invasion, and metastasis in lung cancers. In myoblastoma (MB), overexpression of NRP1 plays a key role in the survival of CSCs, inhibiting the NRP1 via a peptidomimetic specifically targeting the protein by losing their stemness characteristics, thereby decreasing the invasiveness capacity of MB CSCs. Therefore, molecular targeting of NRP1 could prove to be effective as it plays an essential role in tumor invasiveness and mobility of CSCs. Several studies have also mentioned that targeting CSCs can be more effective in increasing survival rates [157] as they are actively involved in metastases and tumor recurrence [166]. More studies are needed to reconcile the critical events in the CSCs in various tumors and study the multiple genes involved, which can be a potential biomarker in the early identification of tumors.

## 12. NRP1 Drug Resistance Mechanism

Precision oncology focuses on administering drugs that specifically target the unique vulnerabilities of individual tumors. Cancer cells are characterized by the activation of oncogene signaling cascades that are exploited for therapy [167]. One of the major challenges is the identification of drug susceptibility and therapeutic approaches to prevent drug resistance by inactivating the underlying molecular mechanisms. NRPs have control over fundamental signaling pathways mediated by tyrosine kinase receptors ranging from tumor cell proliferation and metastatic dissemination to tumor angiogenesis and tumor escape [168]. NRP1, as described in this review, can act as an important biomarker in multiple cancer types, and there is always an elevated expression of NRP1 in cancers. The mechanisms modulating NRP1 expression in cancer cells are still controversial [167], but NRPs serve as predictive biomarkers of drug response. In many cancer studies, a dual-target therapeutic strategy is explored to overcome drug resistance. The resistance mechanism to cetuximab (ctx) in pancreatic ductal adenocarcinoma (PDAC) cells involves the overexpression of active integrin β1, leading to Src-AKT activation and resulting in EGFR ligand-independent proliferation signaling that circumvents the effects of EGFR blockade., a dual-targeting antibody, Ctx-TPP11, by genetically fusing the NRP1-targeting peptide TPP11 to the C terminus of the cetuximab heavy chain was developed that inhibits integrin β1 signaling. A mechanism of sensitization entails the downregulation of active β1 levels via NRP1-coupled internalization facilitated by the TPP11 moiety, resulting in the inhibition of bypass signaling driven by active integrin β1 via Src-AKT hyperactivation leading the resistance mechanism to cetuximab in PDAC [169]. KRAS mutant harboring (KRAS-MUT) non-small cell lung cancer (NSCLC) also has integrin β3-NRP1-KRAS-MUT ternary complex and the downstream signaling of PI3K-AKT signaling leading to primary resistance in vitro and in vivo mouse models. Dual targeting of EGFR and NRP1 with bispecific antibodies has an effective therapeutic strategy for NSCLC [170]. Another study targeted the NRP1/TGB3 axis, a novel mechanism in breast cancer cell line BT-474 correlating high NRP1 with TNC/ITGβ3 signaling but decreases the ABCG2 expression, thereby sensitizing BT-474 NRP1 cells to Adriamycin/cyclophosphamide [171]. Some patient populations initially respond to therapy, but after a few months, they eventually show the progression of the disease. Combination strategies are used to broaden the responding patient population to overcome this challenge of acquired resistance [172]. Sabrina et al. treated melanoma cells with BRAF inhibitors and breast cancer cells with HER2-targeted drugs, which increased the NRP1 expression via JNK-dependent signaling cascade and upregulated the EGFR/IGF1R and mediated adaptive resistance. Combination with NRP1-interfering molecules improved the efficacy of oncogene-targeted drugs [167]. NRPs are pivotal players in tumor angiogenesis, as they bind to members of the VEGF family and interact with tyrosine kinase VEGFR1, VEGFR2, and VEGFR3 [53,173,174]. VEGF-targeted therapies are now approved to treat multiple cancers; current research focuses on identifying biomarkers of responsiveness to anti-VEGF antibodies [80]. Studies using a combination of VEGF inhibitors with NRP1 blocking antibodies to enhance the antiangiogenic effect were conducted, but these failed because of adverse side effects in phase1 patients that assayed MNRP1685A combination with bevacizumab [175].

## 13. Challenges and Future Directions

Despite the promise of targeting NRP1 in cancer therapy, several challenges remain. One significant hurdle is the potential for off-target effects, given the widespread expression of NRP1 in normal tissues. Additionally, the complexity of NRP1 signaling and its interactions with multiple ligands necessitate further research to elucidate its role in different cancer types and stages fully. Future studies will likely focus on refining NRP1-targeted therapies, identifying predictive biomarkers to stratify patient populations, and exploring novel delivery strategies to enhance therapeutic efficacy while minimizing side effects. In conclusion, NRP1 represents a multifaceted target in cancer therapy, influencing angiogenesis, cell migration, invasion, and immune regulation within the tumor microenvironment. As research unravels its complexities, NRP1-targeted therapies promise to improve treatment outcomes and advance the fight against cancer.

### 13.1. NRP1 Omics in Cancer Biology

As of 2022, 1,918,030 new cancer cases and 609,360 cancer deaths are projected to occur in the United States [176]; these numbers can surpass heart diseases as the leading cause of death in the next few years in the USA [176]. Many in vitro and in vivo studies have defined the classical hallmarks of cancer for its invasion and metastasis [177]. New research evidence provides more data on the genes involved in angiogenesis, invasion, and metastasis. The Precision Medicine approach aims to revolutionize the right treatment and medical diagnostics. The treatments have focused on the majority of patients grouped based on types of cancer, the inter-individual differences such as each patient having a different cancer driver mutations or tumor proteomic profiles, but their response to chemotherapy and different prognoses have not been defined previously [178]. A lack of accurate profiling of these patients can lead to sub-optimal choice of treatments. Therefore, Precision Oncology is intended to better identify these inter-individual differences to study disease phenotypes and provide treatment plans based on the patient’s needs [178]. With the advent of Omics and big data analytics, molecular-level information can now be gathered to identify obscure patterns from these data effectively. Thus, the in-depth analysis of genome, epigenome, transcriptome, proteome, and tumor metabolism allows us to dissect tumor biology and heterogeneity [179]. OMICS-based approaches play a pivotal role in understanding the cancer-associated cellular processes and mechanisms by studying the cancer cells at gene, transcript, and protein levels, validating and exploring the interaction networks and their translation from bench to bedside [180]. NRP1 plays a pivotal role in cancer biology through its involvement in various cellular processes, including angiogenesis, immune modulation, and metastasis. The application of omics technologies—genomics, transcriptomics, proteomics, and metabolomics—has significantly advanced our understanding of NRP1’s multifaceted roles in cancer. Here, we provide an overview of how these omics approaches have elucidated the function and regulation of NRP1 in cancer.

#### 13.1.1. Genomics of NRP1 in Cancer

Genomic studies have identified alterations in the NRP1 gene that are associated with cancer. Key findings include gene amplification and mutations. NRP1 gene amplification and mutations have been detected in certain cancers, contributing to aberrant expression and activity. These genetic alterations can increase NRP1 signaling, promoting tumor growth and survival [90]. Single Nucleotide Polymorphisms (SNPs): Specific SNPs in the NRP1 gene have been linked to cancer susceptibility and prognosis. For example, SNPs that affect the regulatory regions of NRP1 may alter its expression levels and impact cancer progression [181].

#### 13.1.2. Transcriptomics of NRP1 in Cancer

Transcriptomic analyses have provided insights into the expression patterns of NRP1 and its downstream targets across different cancers. Differential Expression: RNA sequencing (RNA-seq) studies have shown that NRP1 is often upregulated in various cancers, including breast, prostate, and colorectal cancers. High NRP1 expression is frequently associated with aggressive tumor phenotypes and poor patient outcomes [135].

Splice Variants: Alternative splicing of the NRP1 mRNA can generate different isoforms with distinct functions [182]. Transcriptomic studies have identified specific splice variants that are preferentially expressed in particular cancer types, influencing tumor behavior and response to therapy [183]. Co-expression Networks: Transcriptomics has revealed co-expression networks involving NRP1 and other genes [184]. These networks highlight the pathways and processes that NRP1 may regulate or participate in, providing a broader context for its role in cancer.

#### 13.1.3. Proteomics of NRP1 in Cancer

Proteomic approaches have shed light on the protein expression, interactions, and post-translational modifications of NRP1 in cancer. Protein Expression Profiling: Mass spectrometry-based proteomics has quantified NRP1 protein levels in various cancer tissues and cell lines, confirming its overexpression in many malignancies [90]. Protein–Protein Interactions: Proteomic studies have mapped the interaction partners of NRP1, revealing its involvement in complex signaling networks. These interactions include binding with VEGF receptors, integrins, and other co-receptors, facilitating angiogenesis and tumor cell migration [5]. Post-Translational Modifications: Analysis of NRP1 post-translational modifications, such as phosphorylation and glycosylation, has provided insights into regulating its activity and stability. These modifications can influence NRP1’s function in tumor progression and response to therapy [87].

#### 13.1.4. Metabolomics of NRP1 in Cancer

Metabolomic studies have begun exploring how NRP1 influences cellular metabolism in cancer. Pathways: NRP1 has been implicated in regulating key metabolic pathways that support tumor growth, such as glycolysis and lipid metabolism. Metabolomic profiling has shown that altering NRP1 expression affects the levels of various metabolites, linking NRP1 activity to metabolic reprogramming in cancer cells [136]. Metabolite Biomarkers: Changes in the metabolite profiles associated with NRP1 activity may serve as biomarkers for cancer diagnosis and prognosis. Metabolomics can identify specific metabolic signatures that reflect NRP1’s role in tumor biology [136].

### 13.2. Integrated Omics Approaches

Combining data from genomics, transcriptomics, proteomics, and metabolomics provides a comprehensive view of NRP1’s role in cancer. Integrated omics approaches facilitate the identification of regulatory networks, and understanding how NRP1 interacts with other genes and proteins in regulatory networks helps delineate its role in cancer signaling pathways [14]. Therapeutic Targets: Omics data can identify potential therapeutic targets within the NRP1 signaling axis, offering new avenues for drug development. Biomarkers: Integrated omics can uncover biomarkers for patient stratification, aiding in developing personalized cancer therapies based on NRP1 activity [185]. Omics technologies have significantly advanced our understanding of NRP1’s roles in cancer, from genetic alterations and expression patterns to protein interactions and metabolic effects. These comprehensive insights provide a solid foundation for developing targeted therapies and improving cancer diagnostics and treatment strategies. Continued research using integrated omics approaches will further elucidate the complex roles of NRP1 in cancer biology and therapeutic response. A comprehensive pipeline for bioinformatic analysis and various applications in cancer has been presented in the figure below (Figure 9).

The expression of NRP1 across all cancers and controls in TCGA datasets was plotted using UALCAN, where high and significant expression between normal and tumor was seen in glioblastoma, kidney cancer, and pancreatic cancer; the details are presented in Figure 10A. The expression of NRP1 in control and the GBM, pancreatic cancers, and renal cancers was plotted using a box plot in GEPIA2, where we observed that in tumor conditions, there is a higher expression of NRP1 than in controls. (Figure 10E) Using UALCAN, we extracted the top 100 positively correlated genes [35] and plotted the clustergram for Reactome pathway analysis 2022 in Enrichr [36,186,187], considering the combined score of Z-values and *p*-values. From this clustergram, in GBM we observe that along with TGFβ, many genes such as *ITGB1*, *ITGA2*, *COL5A1*, and *LAMB2* positively correlate with NRP1. These genes play a pivotal in extracellular matrix re organization, TGFβ signaling events, and syndecan interactions, non-integrin membrane interactions (Figure 10B). Whereas, in pancreatic cancer, genes such as *PRKACA*, *CALM1*, *GPS2*, *MECP2* show a positive correlation with NRP1; these genes are involved in upregulation of kinas activity, immune filteration, G-protein pathway suppressor 2 in progression and development of pancreatic (Figure 10C). Positively correlated genes in KIRC include *ZNF 350*, *RING1*, *PARP1*, *SUMO1 HDAC family* involved in RNA polymerase II transcription, Sumoylation of transcription co-factors and chromatic organization and signaling of NOTCH receptors. Further, OMICS-based approaches are required to understand their cellular processes (Figure 10D).

## 14. Conclusions

Research continues to focus on better understanding the complex roles of VEGF and NRP1 in cancer. Ongoing studies aim to identify novel inhibitors with greater specificity and efficacy and elucidate the resistance mechanisms to current therapies. Additionally, exploring biomarkers for predicting response to VEGF and NRP1-targeted treatments is an active area of investigation. A growing body of research shows that NRP1 is a distinct immune modulator in cancer immunotherapy. NRP1, expressed simultaneously with several IRs in the TME, intrinsically controls Treg cell and CD8+ T cell activity to suppress antitumor immunity jointly. By encouraging their recruitment to the tumor bed and maintaining their functional stability amidst continuing inflammation, NRP1 promotes the activity of intratumoral Treg cells. NRP1+ Treg cells are more prevalent in malignancies in cancer patients, and therapeutic intervention is linked to a reduction in NRP1 expression in peripheral Treg cells. Despite having a negligible influence on CD8+ T cell effector activity, NRP1 has a unique effect compared with other IRs on memory formation and lasting response. These unique properties might result in clinical efficacy distinct from current standard-of-care checkpoint inhibitors, offering a sound basis for therapeutic combinations.

In summary, VEGF and NRP1 are integral to cancer pathology, mainly through their roles in promoting angiogenesis, tumor growth, and metastasis. Targeting these molecules holds significant promise for improving cancer treatment and patient outcomes. NRP1 significantly influences tumor development and incidence. It affects tumor immunity, cancer migration, and angiogenesis. Earlier in this paper, some of the NRP1 signaling pathways were mentioned. Several malignancies may respond well to treatment that targets these pathways. However, more research is required to understand how NRP1’s molecular mechanism contributes to the development and spread of cancer. To attain a complete response in cancer patients, inhibitors of NRP1 function must be coupled with other therapeutic modalities, such as immunotherapy, radiation, and chemotherapy.

## Figures and Tables

**Figure 1 curroncol-32-00203-f001:**
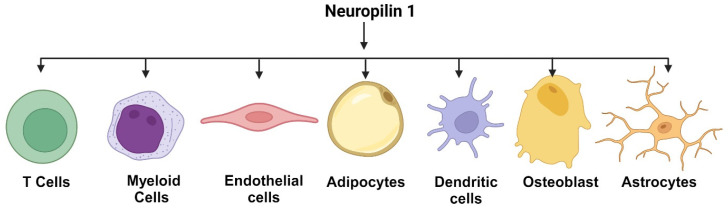
Schematic showing the NRP1 expressions in different cells in humans. The schematics were created using BioRender (https://www.biorender.com/ (accessed on 18 September 2023)).

**Figure 2 curroncol-32-00203-f002:**
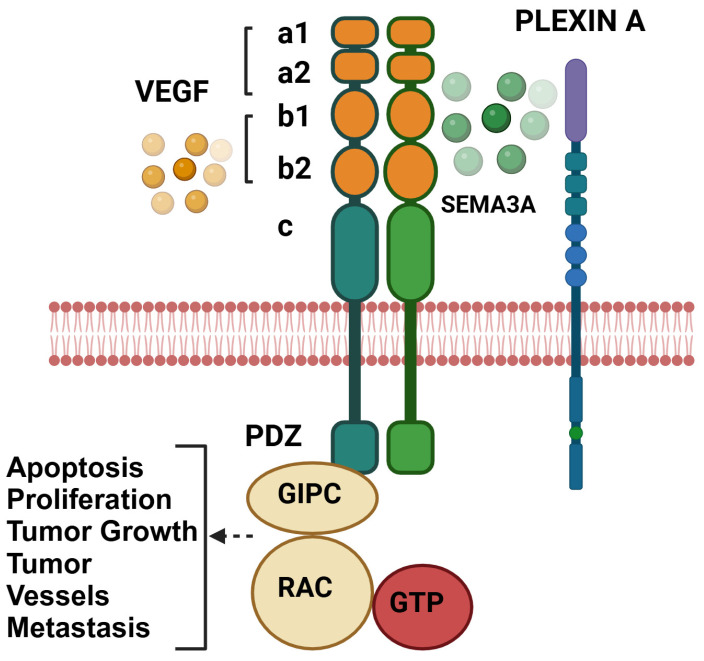
Schematic representation of semaphorins and the functional significance of semaphorins and VEGFA with respect to NRP1. The image was created using BioRender (https://www.biorender.com/). PDZ; Post synaptic density protein (PSD95), Drosophila disc large tumor suppressor (Dlg1), and zonula occludens-1 protein (zo-1). GIPC; GAIP interacting protein, C terminus, RAC; Ras-related C3 botulinum toxin substrate 1, GTP; Guanosine triphosphate. a1,a2,b1,b2,c: Extracellular domains.

**Figure 3 curroncol-32-00203-f003:**
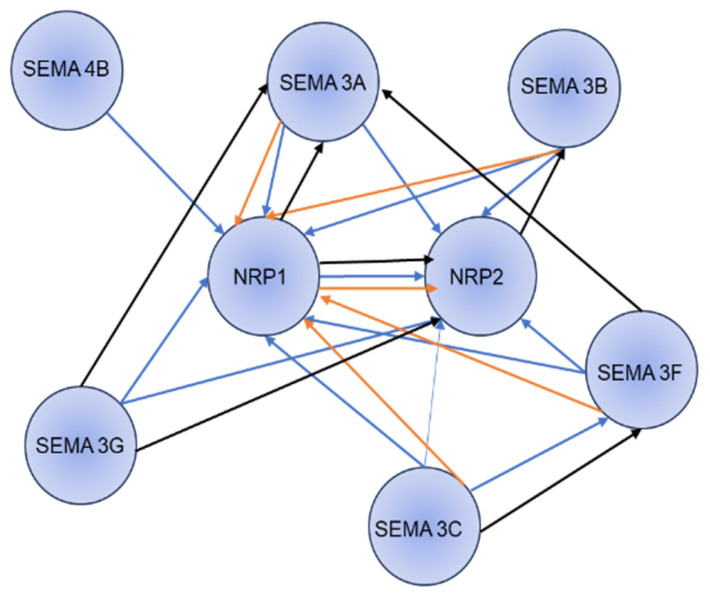
STRING network interaction between NRPs and Sema family. The circles represent protein, and the colored lines represent the interaction between the two genes. Blue-colored lines indicate gene fusions; Black-colored lines indicate co-occurrence; Orange-colored lines indicate gene co-expression between two genes.

**Figure 4 curroncol-32-00203-f004:**
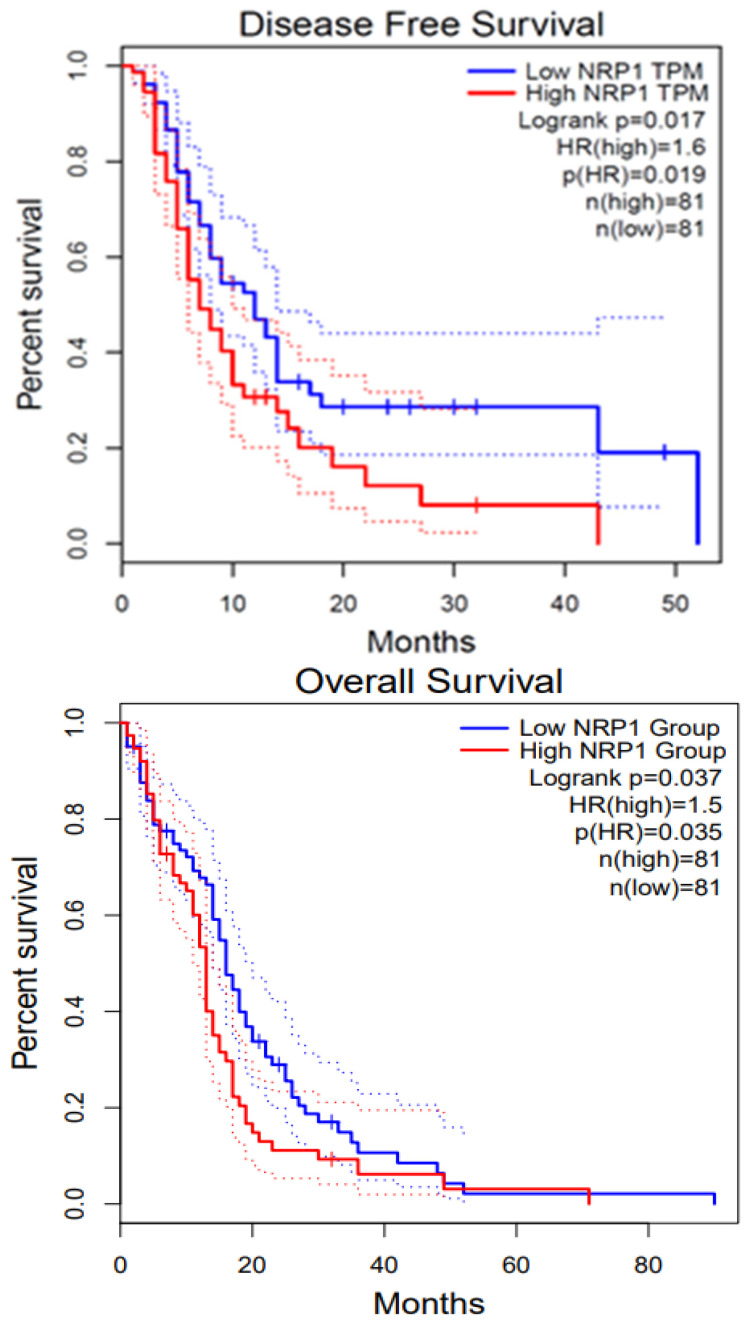
Disease-free survival and overall survival analysis of GBM patients with respect to NRP1. The *X*-axis is given in months, and the *Y*-axis is the survival percentage. The image was extracted from GEPIA2. The solid lines represent the Kaplan-Meier survival estimate for high (Red) or low (Blue) expression of the gene. The dashed lines outline the upper and lower boundaries of the 95% Confidence Interval (CI).

**Figure 5 curroncol-32-00203-f005:**
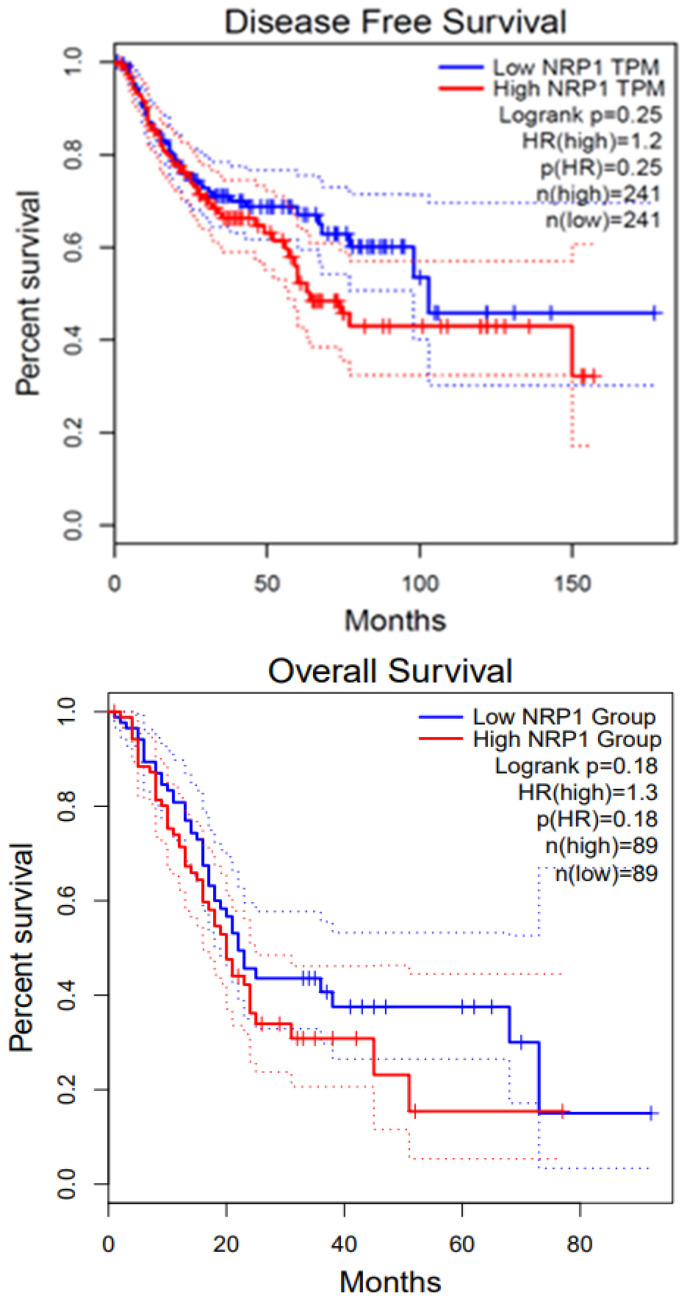
Disease-free survival and Overall survival analysis for PAAD with respect to NRP1. The *X*-axis is given in months, and the *Y*-axis is the percentage of survival. Image extracted from GEPIA2. The solid lines represent the Kaplan-Meier survival estimate for high (Red) or low (Blue) expression of the gene. The dashed lines outline the upper and lower boundaries of the 95% Confidence Interval (CI).

**Figure 6 curroncol-32-00203-f006:**
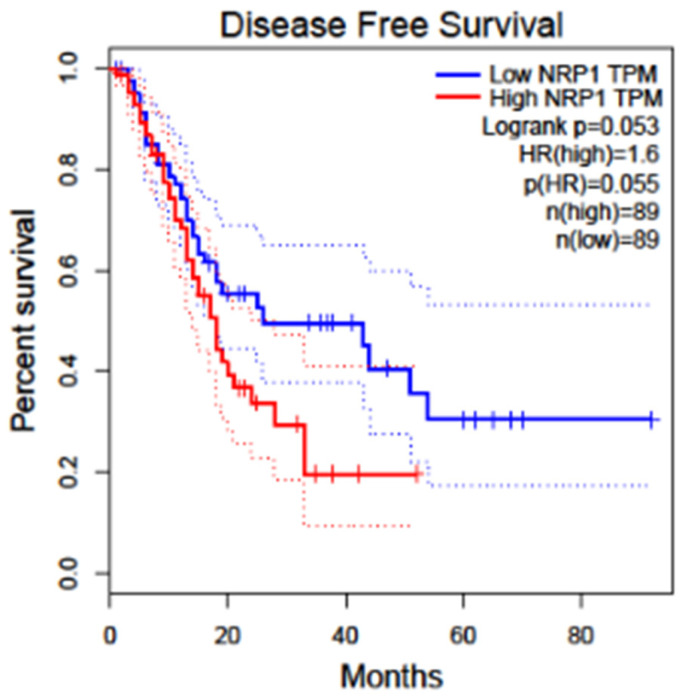
Disease-free survival and Overall survival analysis for KIRC with respect to NRP1. The *X*-axis is given in months, and the *Y*-axis is the survival percentage. Image extracted from GEPIA2. The solid lines represent the Kaplan-Meier survival estimate for high (Red) or low (Blue) expression of the gene. The dashed lines outline the upper and lower boundaries of the 95% Confidence Interval (CI).

**Figure 7 curroncol-32-00203-f007:**
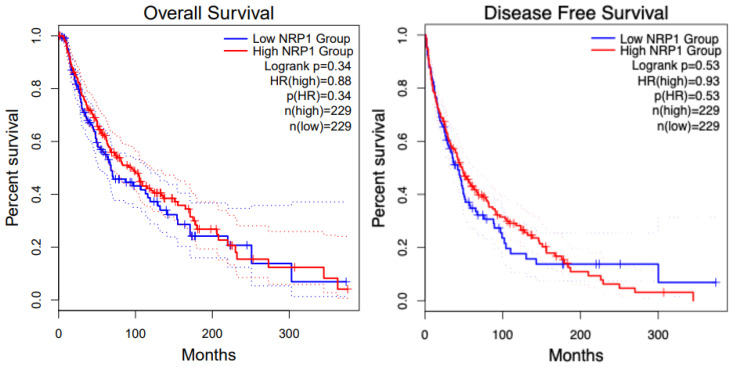
Disease-free survival and Overall survival analysis for SKCM with respect to NRP1. The *X*-axis is given in months, and the *Y*-axis is the survival percentage. Image extracted from GEPIA2. The solid lines represent the Kaplan-Meier survival estimate for high (Red) or low (Blue) expression of the gene. The dashed lines outline the upper and lower boundaries of the 95% Confidence Interval (CI).

**Figure 8 curroncol-32-00203-f008:**
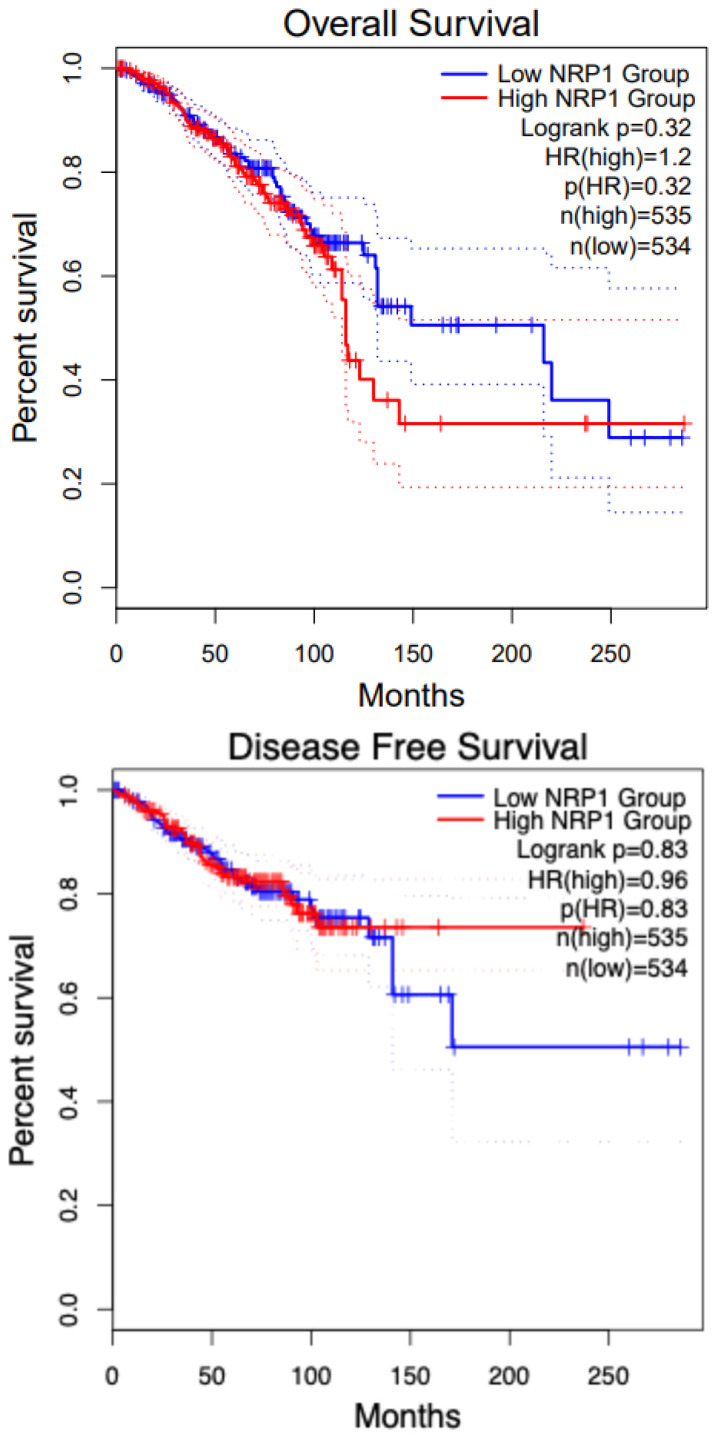
Disease-free survival and Overall survival analysis of BRCA with respect to NRP1. The *X*-axis is given in months, and the *Y*-axis is the survival percentage. Image extracted from GEPIA2. The solid lines represent the Kaplan-Meier survival estimate for high (Red) or low (Blue) expression of the gene. The dashed lines outline the upper and lower boundaries of the 95% Confidence Interval (CI).

**Figure 9 curroncol-32-00203-f009:**
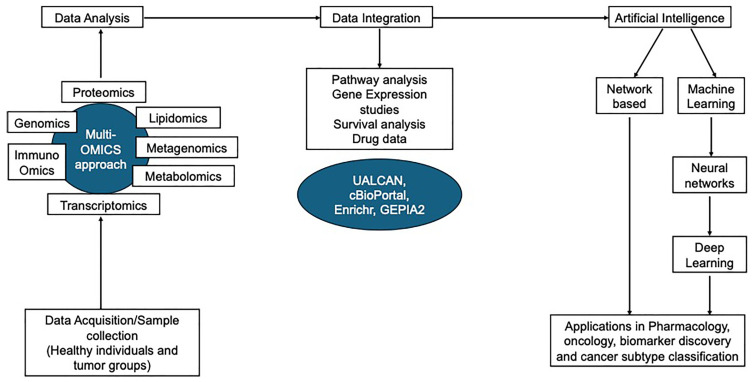
Integrated bioinformatics pipeline involved in various applications of Oncology, cancer subtype classification, and biomarker discovery using various online databases (UALCAN, cBioportal, Enrichr, GEPIA2).

**Figure 10 curroncol-32-00203-f010:**
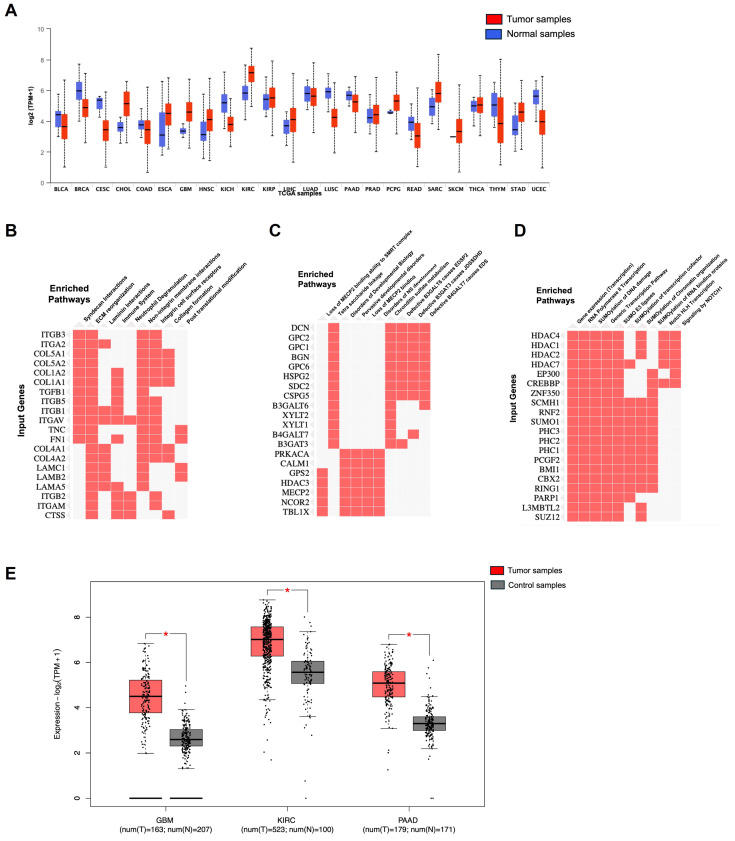
OMICS-based analysis using UALCAN, Enrichr, and GEPIA2 (**A**) Pan-cancer analysis of NRP1 across all cancers in TCGA datasets. Blue indicates normal/control samples, and red indicates tumor. *X*-axis: TCGA datasets for normal and tumor samples. *Y*-axis: log2 scale. (**B**–**D**) Clustergram plotted for reactome pathways 2022 using Enrichr.R package(3.4) Enriched terms: various pathways, input genes: positively correlated genes to NRP1. Glioblastoma, Pancreatic Cancer, Kidney Cancer. (**E**) NRP1 expression in various cancers (GBM, KIRC, PAAD) compared with controls. Red-colored are tumors, and black-colored are controls. *X*-axis: types of cancers, *Y*-axis: log10 (TPM) scale. Astreik represents the statistical significance, *, *p* < 0.05.

**Table 1 curroncol-32-00203-t001:** NRP1 and its interacting domains with extracellular ligands and the downstream signaling.

Ligand	NRP1 Domain	Effect	References
SEMA3A	CUB (a1/a2) extracellular domain	Promotes prolonged T cell–DC interaction and T cell activation and IL-10 secretion	[38,39,40]
SEMA4A	CUB (a1/a2) extracellular domain	Promotes contact-independent Treg function (via IL-10 and IL-35) and maintains Treg stability in vivo	[41]
TGF-β	b1/b2 extracellular domain	Activates latent LAP–TGF-β enhancing TGF-β immune suppression and TGF-β mediated Treg generation.	[42]
VEGF165/145	b1/b2 extracellular domain	VEGF165 enhances VEGFR2–NRP1 complex formation by acting as a ‘bridging molecule’—this enhances the proangiogenic effects of VEGF165	[43,44]
HGF	b1/b2 extracellular domain	NRP1–HGF binding enhances c-Met signaling, promoting endothelial cell proliferation and angiogenesis	[45,46]
PDGF	Unconfirmed physical interaction with NRP1; possibly b1/b2 domain	PDGF upregulates NRP1 expression, promoting VSMC mobilization and angiogenesis	[47]
FGF2	Unconfirmed physical interaction with NRP1; possibly b1/b2 domain	NRP1 binding of FGF2 enhances the FGF2 growth stimulatory functions and proangiogenic activity	[48]
PIGF	b1/b2 extracellular domain	PlGF signals through its receptor, NRP1, promoting angiogenesis and tumor growth	[49]

**Table 2 curroncol-32-00203-t002:** NRPs and their function in various immune cells.

Neuropilins Role in Immune Cells			
Immune Cells	Neuropilin Type	Function	References
Dendritic cells	NRP1	Mediates Primary IR activation by antigen processing and presentation by DCs	[39]
	NRP2	Differentiation from Monocytes to DCs, protecting their migration by sialyation where DC activates T-cells	[112,113]
Macrophages	NRP1	Promotes immune suppressive role and induces a protumoral response	[114]
	NRP2	Differentiation of Monocytes to macrophages to induce phagocytosis, NRP2 sialyation reduces phagocytosis capacity	[39,115]
		T cells	
Cytotoxic T cells	NRP1	Promotes antigen recognition, a biomarker to determine the efficacy of anti-PD-1 immunotherapies	[26,107]
Helper T cells	NRP1	On CD4+, T cells promote B cell differentiation	[26,102,116]
	NRP1	On Treg cells and CD4+ T cells induce Immunosuppressive function	
NKT cells	NRP1/NRP2	Unknown	[26]
T regulatory cells	NRP1	Attracts to VEGF where NRP1 acts as a co-receptor enhancing infiltration of tumors and Immunosuppressive response	[117]
	NRP2	Interaction between NRP2, SEMA3A, PlexinA1 inhibit immature T cell migration	[118]

## Data Availability

Data supporting reported results can be found, at UALCAN (https://ualcan.path.uab.edu), Enrichr (https://maayanlab.cloud/Enrichr/), GEPIA2 (http://gepia2.cancer-pku.cn/#index), STRING (https://string-db.org) accessed on 22 March 2025.

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
