# Peer review of "Neuropilin-1: A Multifaceted Target for Cancer Therapy"

_curroncol, 2025, doi:10.3390/curroncol32040203_

Round 1
Reviewer 1 Report
Comments and Suggestions for Authors
The work of R.S.Angom et al. is a systematic review providing information on the properties of neuropilin-1 (NRP1) and its involvement in physiology and pathophysiology (with emphasis on carcinogenesis).
The authors address the following issues.
- NRP1 interaction with extracellular ligands (semaphorins, VEGFs) and physiological consequences of these interactions, primarily in the stimulation of angiogenesis.
- Participation of NRP1 in tumor growing and metastasis, which is mainly due to its angiogenic effects.
- Regulation of immune cell activity by NRP1, which is essential in the development of tumors.
- Role of NRP1 in different types of cancer (glioblastoma, pancreatic cancer, renal cancer, melanoma, breast cancer).
- Different types of NRP1-targeting cancer therapy are reviewed; the difficulties of targeting therapy due to the wide distribution of NRP in body tissues are also discussed.
- The omics logistics of NRP1 in cancer pathology are also considered.
The work impresses with its logical and comprehensible presentation and is certainly useful for the reader concerning with the problem.
The reviewer has no major comments on this paper
Minor comments.
1. Authors should refer to Figure 2 in the text. Abbreviations PDZ, GIPC, RAC and GTP should be deciphered in legend to this figure. If Plexin A is represented as receptor for Semaphorin, then please, indicate it graphically.
2. Authors need to mention figure 3 in the text of manuscript. Which program was used for creation of figure 3? Please, refer to it.
3. Tables 1-3 should not contain “shows” in the title. Just a title only.
4. Some abbreviation should be deciphered: PAE cells (Line 250), GBM (lines 383, 392) and so on.
5. The sentences in lines 175-177 and 187-189 are incomprehensible.
6. The text contains some minor unchecked typos and needs to be carefully re-read.
Author Response
We thank the reviewers for all the comments and valuable suggestions. The reviewers' comments have significantly improved our manuscript. In this revision, we have carefully addressed all the comments suggested by the reviewers. Please see our point-by-point response below. Response to Reviewer 1: 1. Authors should refer to Figure 2 in the text. Abbreviations PDZ, GIPC, RAC and GTP should be deciphered in legend to this figure. If Plexin A is represented as receptor for Semaphorins, then please, indicate it graphically: Response: Thank you for these suggestions. The abbreviations have been added to the figure legend. Please see the changes as mentioned below: The abbreviations have been added to the legend in Figure 2 at lines 147-150. PDZ- post synaptic density protein (PSD95), Drosophila disc large tumor suppressor (Dlg1), and zonula occludens-1 protein (zo-1). GIPC; GAIP interacting protein, C terminus. RAC; Ras-related C3 botulinum toxin substrate 1, GTP; Guanosine triphosphate. We confirmed that the previous manuscript already referred to Figure 2 in line no. 84. 2. Authors need to mention figure 3 in the text of manuscript. Which program was used for creation of figure 3? Please, refer to it. Response: We have now updated the manuscript and mentioned Figure 3 in line 224. Please see the highlighted text. We used the STRING database to generate Figure 3. Please see the reference and the link below. Szklarczyk D, Kirsch R, Koutrouli M, Nastou K, Mehryary F, Hachilif R, Annika GL, Fang T, Doncheva NT, Pyysalo S, Bork P, Jensen LJ, von Mering C. 'The STRING database in 2023: protein-protein association networks and functional enrichment analyses for any sequenced genome of interest.' Nucleic Acids Res. 2023 Jan 6;51(D1): D638-646. 3. Tables 1-3 should not contain “shows” in the title. Just a title only. Response: We have now updated the manuscript and highlighted the new titles in lines 115 for Table 1 and line 483 for Table 2. 4. Some abbreviations should be deciphered: PAE cells (Line 250), GBM (lines 383, 392), and so on. Response: We have now added the abbreviations at their respective places. Line 487, GBM abbreviation is added - Glioblastoma (GBM). A list of glossary terms has been provided for a clearer understanding of complex terms at the end of the manuscript. 5. The sentences in lines 175-177 and 187-189 are incomprehensible. Response: We have completely removed the chapter with the title “anti-angiogenic” because it focuses more on semaphorins, and our primary goal is to discuss NRP1. Please see the updated Chapter 3, “Semaphorins-NRP1 signaling,” which provides more information in the manuscript in Lines 201-253. 6. The text contains some minor unchecked typos and needs to be carefully re-read- Response: Typos have been addressed throughout the text.
Reviewer 2 Report
Comments and Suggestions for Authors
This Review article by Angom and coworkers aims to provide a comprehensive survey of the multifaceted involvement of Neuropilin-1 (NRP1) in cancer, especially focusing on its relevance as therapeutic target. The work is very broad, aiming to cover multiple aspects of the topic; however, the abundance of information is not matched by clear and efficacious discussion, making the message at times repetitive, in other cases incomplete, or unbalanced, or even potentially confusing. References to discussed papers are often missing (or inappropriately cited). English language, surprisingly, needs extensive revision. Representative examples of these issues are listed below:
Line 48: the text refers to NRP1 interacting “with multiple ligands and co-receptors”, however only few soluble ligands are mentioned here and no example of co-receptor, which gives an incomplete picture of the molecule in this introduction. This information should be integrated.
Lines 88-94 describe NRP1-GAG interactions in detail; however, no appropriate reference to research literature is currently cited.
Conversely to what mentioned above, Table 1 is supposed to list “NRP1 interactions with extracellular ligands”, while instead a large part of it concerns transmembrane proteins, mostly signaling receptors. This concept should be clarified.
Lines 119-121 specifically describe the impact of NRP1 genetic deficiency in adult mice; however, the associated reference is an older review article instead of a research article demonstrating these findings.
Lines 121-124 mention the link between NRP1 and TGF-beta signaling, however a reference to pertinent literature is missing.
Line 122 also needs English revision: “The pro/anti-functioning of NRP1 in cancer shows a novel pathway binding of KRAS and TGF-beta signaling through NRP1….”
Chapter 2 is entitled “Semaphorins-NRP-1 signalling:”, however the discussed information do not give a clear and focused picture of our knowledge about NRP1-Semaphorin interactions. It should be specified that NRP1 is mainly known to interact with class 3 soluble semaphorins and an accurate list of known ligands and associated functions should be provided. Instead, the Authors here describe the general structure of semaphorins, indulging in details concerning members of other classes that have never been reported to interact with NRP1. Table 2, furthermore, includes a full list of all semaphorin family members, most of which do not signal via Neuropilins and rather bind to Plexin receptors, which are barely mentioned in this article, while are the major effectors of NRP1 function in several contexts. Actually, in the table, the relevance of NRP1 in semaphorin signaling is not even mentioned, which is really surprising considering the focus of this article. Only the last statement in this chapter (line 168) swiftly alludes to the fundamental information which should have been discussed in depth here.
Line 164 refers to “cleavage in the basic domain which potentiated the activity of sema3A”. This statement, applied to Sema3A, is inconsistent with the largest body of evidence in literature and, notably, is not associated with any cited reference. The authors should amend the statement or cite the appropriate literature.
The whole Table 2 lacks appropriate references for each of the illustrated genes and molecules. In the table caption are merely indicated 3 older review articles (dated in 2017/2018), suggesting that the table does not report an updated picture of the knowledge in the field.
Chapter 3 is surprisingly entitled “Anti-angiogenic function:”, being unclear the involvement of NRP1 in this concept. Surely NRP1 is not considered an anti-angiogenic factor, although some of its ligands may elicit this effect, so this title should be revised. Actually, assessing this chapter was particularly difficult, due to poor English language and confusing mixup of diverse concepts. It should be completely revised, clarifying the functional role of NRP1 in angiogenesis, which mostly involves its interaction with VEGF and KDR, and then the regulatory role of a subset of semaphorins, some of which are considered anti-angiogenic. Since the subsequent Chapter 4 is focused on VEGFA and NRP1, I recommend swapping the two, to improve the clarity of the elaborated concepts.
Lines 248-250 contain a confusing statement, which should be revised: “NRP1 contributes to VEGF-induced vascular hyperpermeability in in vitro experiments of transendothelial electrical resistance of PAE cells co-infected with NRP1 and VEGFR2 after VEGF165 stimulation”. Too many experimental details and unclear message. Moreover, importantly, no cited reference. Then, the text follows “But no known mechanism exists to explain this, and this has not been replicated. Also, this study’s cell lines were not endothelial cells”. The Authors can do better discussing this study in the frame of current literature. In the subsequent lines, various other findings are mentioned never including appropriate references to the publications.
Chapter 5 is entitled “NRP1 promotes tumor proliferation and migration” and one would expect this to be one of the key parts of the article. Instead, are briefly mentioned here just a few scattered studies, without connections and insightful discussion, which provides a far-from-complete picture of the broad and relevant literature on the topic. The chapter needs profound revision, aimed at providing mechanistic analysis of the reported mechanisms and functions of NRP1 in controlling cancer cell proliferation and migration. A subsequent Chapter 7 is entitled “Role of NRP1 in Cancer”. The Authors should seriously consider to move this chapter earlier in the article, providing the frame for a general introduction to the topic, followed by detailed mechanistic analysis of its role in the regulation of cancer cells, tumor vasculature and immune cells.
Lines 273-275, notably, contain this confusing statement: “Additionally, overexpression of PDGF and its receptor (PDGFR), angiogenic factors are strongly linked to carcinogenesis and progression (67,68), which is a classical character of many malignancies”. One is tempted to imagine that two independent pieces of text have been pasted together… Moreover, defining carcinogenesis and progression as a classical character of many malignances sounds weird.
Chapter 6, entitled “Role of NRP1 in immune cells” starts off with a long paragraph (lines 289-311), which is mostly unclear and unfocused, with respect to the topic of the chapter. It entirely serves to discuss a single previous publication of the authors’ group (ref. 76), concerning signaling in brain microvascular endothelial cells. At least, this part should be placed more appropriately in the article. The subsequent paragraph (lines 312-331) is -conversely- completely devoid of references to discussed literature.
Line 335 needs English revision.
Lines 451-456 need English revision.
Subchapter 13.1 deals with NRP1-targeting approaches in cancer therapy without providing a single reference for the various mentioned studies and validated approaches. Although the latter are further discussed in the subsequent chapters, key references should be added also in the introductory part. Moreover, it is surprising that the Authors are not mentioning in their review the relevance of NRP1-targeting approaches aimed at interfering with drug-resistance mechanisms in cancer therapy, although this topic is well covered in literature (e.g. PMID: 27797376, PMID: 29728077, PMID: 29953416, PMID: 31521695, PMID: 31027288, PMID: 32784465, PMID: 33128042, PMID: 36550208, PMID: 38448392, PMID: 39226661, PMID: 38960889, etc.); with the notable exception of cited Ref. 97, which is a paper published by the same authors.
Line 528 calls out Figure 2, but this is not actually linked to what discussed in the text. Moreover, the statement at lines 531-533 should be rephrased for clarity.
The whole subchapter 13.4 and chapter 14 need English revision.
Line 631 is wrong, where it reads “Deletions of various regions in the NRP1 gene in macrophages….”, since this study analyzed cell type-specific conditional NRP1-KO mice.
Chapter 15 is entitled “Challenges and Future Directions”. Yet, this is surprisingly not placed at the conclusion of the article. In fact, additional Chapters 16-21 concern various aspects, mainly concerning “omic” perspectives…. and then another “Conclusions” chapter no. 22 can be found… If chapter 15 only refers to conclusions of the “NRP1-targeting” topic, then it should be numbered as subchapter 13.5.
Lines 744-750: with reference to NRP1 Splice variants and Co-expression Networks, there is no mention of specific conclusions of the generically mentioned studies, not cited references to access the published data. These parts should be integrated.
Comments on the Quality of English LanguageEnglish language needs extensive revision.
Author Response
Response to Reviewer 2:
We thank the reviewers for all the comments and valuable suggestions. The reviewers' comments have significantly improved our manuscript. In this revision, we have carefully addressed all the comments suggested by the reviewers. Please see our point-by-point response below.
This Review article by Angom and coworkers aims to provide a comprehensive survey of the multifaceted involvement of Neuropilin-1 (NRP1) in cancer, especially focusing on its relevance as therapeutic target. The work is very broad, aiming to cover multiple aspects of the topic; however, the abundance of information is not matched by clear and efficacious discussion, making the message at times repetitive, in other cases incomplete, or unbalanced, or even potentially confusing. References to discussed papers are often missing (or inappropriately cited). English language, surprisingly, needs extensive revision. Representative examples of these issues are listed below:
Response: We appreciate the reviewer’s comments. To address these concerns, we have now updated most of the sections in our manuscript and included more references. Please see our point-by-point response below.
Line 48: the text refers to NRP1 interacting “with multiple ligands and co-receptors”, however only few soluble ligands are mentioned here and no example of co-receptor, which gives an incomplete picture of the molecule in this introduction. This information should be integrated.
Response: We have updated the correct sentence in the manuscript. The sentence has been revised to “NRP1 interacts with multiple ligands, such as vascular endothelial growth factor (VEGF), semaphorins, and transforming growth factor-beta (TGF-β), mediating diverse signaling pathways that promote oncogenesis (6)”. This change is highlighted in text from lines 49-53.
Lines 88-94 describe NRP1-GAG interactions in detail; however, no appropriate reference to research literature is currently cited.
Response: Thank you for your suggestion. We have updated the manuscript with the related references, PMID:16763549 and PMID: 29067024, at lines 98 and 101.
Conversely to what mentioned above, Table 1 is supposed to list “NRP1 interactions with extracellular ligands”, while instead a large part of it concerns transmembrane proteins, mostly signaling receptors. This concept should be clarified.
Response: We appreciate the reviewer’s suggestions; to convey the proper information we have now modified the Title of the Table1 as “NRP1 and its interacting domains with extracellular ligands and the downstream signaling. Additionally, we have now focused our Table 1 with NRP1 and its interaction with various ligands. Please see the update to Table 1 in line 115.
Lines 119-121 specifically describe the impact of NRP1 genetic deficiency in adult mice; however, the associated reference is an older review article instead of a research article demonstrating these findings.
Response: The manuscript is now updated with the references PMID: 32210960, PMID: 18180379, and PMID:33876574 at lines: 127, 128 and 131. Please find the highlighted text in the manuscript.
Lines 121-124 mention the link between NRP1 and TGFb- signaling, however a reference to pertinent literature is missing.
Response: Thank you for your suggestion. We updated the manuscript with the respective reference PMID: 28938007 at Line 131. The link between TGFb and NRP1 is discussed in chapter 8.1 GBM and NRP1, at line no 494-501.
Line 122 also needs English revision: “The pro/anti-functioning of NRP1 in cancer shows a novel pathway binding of KRAS and TGF-beta signaling through NRP1….”
Response: Sentence has been changed to “The pro/anti-angiogenic function of NRP1 shows a novel pathway in cancer metastasis mediated by KRAS and TGF-β signaling through NRP1.” at line 129-131 in the manuscript.
Chapter 2 is entitled “Semaphorins-NRP-1 signaling:”, however the discussed information do not give a clear and focused picture of our knowledge about NRP1-Semaphorin interactions. It should be specified that NRP1 is mainly known to interact with class 3 soluble semaphorins and an accurate list of known ligands and associated functions should be provided. Instead, the Authors here describe the general structure of semaphorins, indulging in detail concerning members of other classes that have never been reported to interact with NRP1.
Response: Thank you for your suggestion. Lines describing the in-depth details of semaphorin structure are eliminated from the manuscript. Information regarding Semaphorins and NRP1 are added in lines 212-253. Please see the text under the new Chapter 3, “Semaphorins and NRP1 signaling.”
Table 2, furthermore, includes a full list of all semaphorin family members, most of which do not signal via Neuropilins and rather bind to Plexin receptors, which are barely mentioned in this article, while are the major effectors of NRP1 function in several contexts. In the table, the relevance of NRP1 in semaphorin signaling is not even mentioned, which is really surprising considering the focus of this article. Only the last statement in this chapter (line 168) swiftly alludes to the fundamental information which should have been discussed in depth here.
Response: Thank you for your suggestion. Table 2: Classes of semaphorins and domains, existence, occurrence, and their physiological function in humans, focuses more on Sema family and therefore we excluded the table2 from the manuscript as the major focus of the manuscript is NRP1 and its role in cancers. We also updated the manuscript with neuropilins and plexin signaling. Please refer to the new Chapter 3 in the manuscript for highlighted changes.
Line 164 refers to “cleavage in the basic domain which potentiated the activity of sema3A”. This statement, applied to Sema3A, is inconsistent with the largest body of evidence in literature and, notably, is not associated with any cited reference. The authors should amend the statement or cite the appropriate literature.
Response: These lines are completely removed from the manuscript as it discusses in-depth about semaphorins whereas our primary focus is on NRP1. Chapter specific to NRP1 signaling with Semaphorins has been included. Please refer to highlighted text in new chapter 3.
The whole Table 2 lacks appropriate references for each of the illustrated genes and molecules. In the table caption are merely indicated 3 older review articles (dated in 2017/2018), suggesting that the table does not report an updated picture of the knowledge in the field.
Response: Thank you for your suggestion. As Table 2 discussed various types of semaphorin families and the physiological role of semaphorins, we excluded this from our current manuscript to maintain the focus of NRP1 signaling in cancers.
Chapter 3 is surprisingly entitled “Anti-angiogenic function:”, being unclear the involvement of NRP1 in this concept. Surely NRP1 is not considered an anti-angiogenic factor, although some of its ligands may elicit this effect, so this title should be revised. Actually, assessing this chapter was particularly difficult, due to poor English language and confusing mixup of diverse concepts. It should be completely revised, clarifying the functional role of NRP1 in angiogenesis, which mostly involves its interaction with VEGF and KDR, and then the regulatory role of a subset of semaphorins, some of which are considered anti-angiogenic.
Response: Thank you for this correction and the kind suggestions. We have updated the information by clarifying the functional role of NRP1 in angiogenesis, which mostly involves its interaction with VEGF and KDR, and then the regulatory role of a subset of semaphorins, some of which are considered anti-angiogenic. Please see text line nos. 312-381 under Chapter 5.
Since the subsequent Chapter 4 is focused on VEGFA and NRP1, I recommend swapping the two, to improve the clarity of the elaborated concepts.
Response: We appreciate the reviewer’s suggestion. We have accordingly restructured the paragraph to reflect the correct information and the continuity of the sentence. Please see line no 141-193 and renumbered this to new chapter 2.
Lines 248-250 contain a confusing statement, which should be revised: “NRP1 contributes to VEGF-induced vascular hyperpermeability in in vitro experiments of trans endothelial electrical resistance of PAE cells co-infected with NRP1 and VEGFR2 after VEGF165 stimulation”. Too many experimental details and unclear message. Moreover, importantly, no cited reference. Then, the text follows “But no known mechanism exists to explain this, and this has not been replicated. Also, this study’s cell lines were not endothelial cells”. The Authors can do better discussing this study in the frame of current literature. In the subsequent lines, various other findings are mentioned never including appropriate references to the publications.
Response: We have changed this statement and included a new reference in the manuscript. Please refer to lines 190-193 under the new Chapter 2.
Chapter 5 is entitled “NRP1 promotes tumor proliferation and migration” and one would expect this to be one of the key parts of the article. Instead, are briefly mentioned here just a few scattered studies, without connections and insightful discussion, which provides a far-from-complete picture of the broad and relevant literature on the topic. The chapter needs profound revision, aimed at providing mechanistic analysis of the reported mechanisms and functions of NRP1 in controlling cancer cell proliferation and migration.
Response: Thank you for your insightful suggestion. Please find the compiled new chapter 5 titled “NRP1 angiogenesis, tumor proliferation and migration”. This chapter provides mechanistic analysis of NRP1 mechanisms and function in NRP1 tumor proliferation and migration. Please refer to lines 312-381.
A subsequent Chapter 7 is entitled “Role of NRP1 in Cancer”. The Authors should seriously consider moving this chapter earlier in the article, providing the frame for a general introduction to the topic, followed by detailed mechanistic analysis of its role in the regulation of cancer cells, tumor vasculature and immune cells.
Response: Thank you for your suggestion. The chapter “Role of NRP1 in cancer” has been moved to Chapter 4. The role of NRP1 in angiogenesis, tumor proliferation, and migration is discussed in the new Chapter 5. Please see the updated chapter highlighted in the manuscript.
Lines 273-275, notably, contain this confusing statement: “Additionally, overexpression of PDGF and its receptor (PDGFR), angiogenic factors are strongly linked to carcinogenesis and progression (67,68), which is a classical character of many malignancies”. One is tempted to imagine that two independent pieces of text have been pasted together… Moreover, defining carcinogenesis and progression as a classical character of many malignances sounds weird.
Response: Thank you for your suggestion. As suggested, we have replaced this with a modified paragraph is provided. For more information, please refer to the new Chapter 5 at lines 312-381.
Chapter 6, entitled “Role of NRP1 in immune cells” starts off with a long paragraph (lines 289-311), which is mostly unclear and unfocused, with respect to the topic of the chapter. It entirely serves to discuss a single previous publication of the authors’ group (ref. 76), concerning signaling in brain microvascular endothelial cells. At least, this part should be placed more appropriately in the article. The subsequent paragraph (lines 312-331) is -conversely- completely devoid of references to discussed literature.
Response: Thank you for your kind suggestions. We have completely removed this section as they talk more about signaling in brain. Literature for missing lines has been updated in the manuscript and a revised paragraph of the role of NRP1 in immune cells has been updated in line no. 423-479 .
Line 335 needs English revision.
Response: The s entence has been revised and updated in the manuscript.
Lines 451-456 need English revision.
Response: This sentence has been revised and updated in the manuscript.
Subchapter 13.1 deals with NRP1-targeting approaches in cancer therapy without providing a single reference for the various mentioned studies and validated approaches. Although the latter are further discussed in the subsequent chapters, key references should be added also in the introductory part.
Response: Thank you for the suggestion. This has been updated to New Chapter 9 in the manuscript, and additional references have been updated as highlighted in the new chapter 9. Please see new references 5, 79, 81, 140.
Moreover, it is surprising that the Authors are not mentioning in their review the relevance of NRP1-targeting approaches aimed at interfering with drug-resistance mechanisms in cancer therapy, although this topic is well covered in literature (e.g. PMID: 27797376, PMID: 29728077, PMID: 29953416, PMID: 31521695, PMID: 31027288, PMID: 32784465, PMID: 33128042, PMID: 36550208, PMID: 38448392, PMID: 39226661, PMID: 38960889, etc.); with the notable exception of cited Ref. 97, which is a paper published by the same authors.
Response: Thank you for the suggestion. A new “Chapter 12. NRP1- Drug Resistance Mechanisms” has been added to the manuscript. Please find the highlighted title in the manuscript at lines 803-845.
Line 528 calls out Figure 2, but this is not actually linked to what discussed in the text.
Response: This sentence has been modified and updated in the manuscript. Please find the highlighted text in the line no. 642-643.
Moreover, the statement at lines 531-533 should be rephrased for clarity.
Response: This sentence has been updated in the manuscript please refer to lines 646-648.
The whole subchapter 13.4 and chapter 14 need English revision.
Response: Thank you for your suggestions. We have revised these chapters. Please see the updated new chapters 10 and 11 in the manuscript.
Line 631 is wrong, where it reads “Deletions of various regions in the NRP1 gene in macrophages….”, since this study analyzed cell type-specific conditional NRP1-KO mice.
Response: The Sentence has been corrected and updated in the manuscript. Please find the highlighted text in line 747-749.
Chapter 15 is entitled “Challenges and Future Directions”. Yet, this is surprisingly not placed at the conclusion of the article. In fact, additional Chapters 16-21 concern various aspects, mainly concerning “omic” perspectives…. and then another “Conclusions” chapter no. 22 can be found… If chapter 15 only refers to conclusions of the “NRP1-targeting” topic, then it should be numbered as subchapter 13.5.
Response: Thank you for your suggestion. New chapter 13 challenges and future directions (at line 840) are subdivided into 2 chapters, 13.1. NRP1 omics in cancer biology and 13.2. Integrated omics approaches. Chapter 14 describes the conclusions. Please find the highlighted changes.
Lines 744-750: with reference to NRP1 Splice variants and Co-expression Networks, there is no mention of specific conclusions of the generically mentioned studies, not cited references to access the published data. These parts should be integrated.
Response: Thank you for your suggestion. We have updated the citations in this paragraph. Please find the highlighted line no 900-906.

Reviewer 3 Report
Comments and Suggestions for Authors
The article titled "Neuropilin-1: A Multifaceted Target for Cancer Therapy
" was reviewed. This article is acceptable provided that it is improved and some errors are fixed.
1- Reduce the percentage of similarity and plagiarism.
2- The article has grammatical errors, so improve it in terms of grammar.
3- The explanations related to the figures are not very clear. It is necessary to explain it more clearly.
4- It is necessary to search for related articles again and add recent studies during the revision stage of the article.
Comments on the Quality of English LanguageThe article titled "Neuropilin-1: A Multifaceted Target for Cancer Therapy
" was reviewed. This article is acceptable provided that it is improved and some errors are fixed.
1- Reduce the percentage of similarity and plagiarism.
2- The article has grammatical errors, so improve it in terms of grammar.
3- The explanations related to the figures are not very clear. It is necessary to explain it more clearly.
4- It is necessary to search for related articles again and add recent studies during the revision stage of the article.
Author Response
Response to Reviewer 3:
We thank the reviewers for all the comments and valuable suggestions. The reviewers' comments have significantly improved our manuscript. In this revision, we have carefully addressed all the comments suggested by the reviewers. Please see our point-by-point response below.
Reduce the percentage of similarity and plagiarism.
Response: Thank you. We have revised the manuscript throughout.
The article has grammatical errors, so improve it in terms of grammar.
Response: Thank You. We have edited the manuscript and corrected the grammar.
The explanations related to the figures are not very clear. It is necessary to explain it more clearly.
Response: Thank you for your suggestions. All the figures have been updated with proper legends. Please find them highlighted in the text.
It is necessary to search for related articles again and add recent studies during the revision stage of the article.
Response: Thank you for your suggestion. The literature has been updated throughout the manuscript.

Reviewer 4 Report
Comments and Suggestions for Authors
The manuscript entitled “Neuropilin-1: A Multifaceted Target for Cancer Therapy” has many mistakes, authors need to rectify many portions.
1. The sentence like "NRP1 is a versatile and multifunctional molecule that has gained significant attention as a potential target for cancer therapy" are repeated unnecessarily. Please avoid reiterating the same ideas multiple times to maintain the reader's engagement.
2. Cross-check and provide brief explanations or a glossary for complex terms.
3. Use shorter, more focused sentences to improve clarity. E.g., "This receptor plays a crucial role in various cellular processes..." makes the text difficult to follow.
4. Claims such as "NRP1 mediates VEGF extracellular migration" lack specific supporting evidence or explanation. Include more data or examples to substantiate claims.
5. A summary of various ligands binding with NRP-1 has been described in Table 1" is mentioned, but the table's key points are not summarized. Provide a concise summary of the table to guide the reader.
6. Please check the text frequently references studies (e.g., Sema3A competing with VEGFR2 for VEGF binding, cleavage sites influencing activity). Are these studies cited appropriately with more recent or pivotal references?
7. Some examples of semaphorins as tumor suppressors (e.g., Sema3B in lung cancer) are given, but others (like Sema3G) are briefly mentioned. Should their roles be elaborated for a more balanced discussion?
8. Elaborate on the physiological and pathological significance of VEGFA isoforms in angiogenesis and tumor biology.
9. Ensure consistent use of terms like semaphorins, Sema proteins, and NRP1 throughout the text.
Good Luck!
Comments on the Quality of English LanguageLanguage correction required!
Author Response
Response to Reviewer 4:
We thank the reviewers for all the comments and valuable suggestions. The reviewers' comments have significantly improved our manuscript. In this revision, we have carefully addressed all the comments suggested by the reviewers. Please see our point-by-point response below.
The sentence like "NRP1 is a versatile and multifunctional molecule that has gained significant attention as a potential target for cancer therapy" are repeated unnecessarily. Please avoid reiterating the same ideas multiple times to maintain the reader's engagement.
Response: Thank you for your suggestion. The repeated sentence has been deleted from multiple locations.
Cross-check and provide brief explanations or a glossary for complex terms.
Response: Thank you for your suggestion. A glossary for complex terms is added at the end of the manuscript in Lines 1005-1052,
Use shorter, more focused sentences to improve clarity. E.g., "This receptor plays a crucial role in various cellular processes..." makes the text difficult to follow.
Response: Thank you for the suggestion. These have been corrected throughout the manuscript.
Claims such as "NRP1 mediates VEGF extracellular migration" lack specific supporting evidence or explanation. Include more data or examples to substantiate claims.
Response: Thank you for your suggestion. These lines have been updated with latest references in the manuscript. Please find the highlighted text at lines 121.
A summary of various ligands binding with NRP-1 has been described in Table 1" is mentioned, but the table's key points are not summarized. Provide a concise summary of the table to guide the reader.
Response: Thank you for your suggestion. We have added this summary, which explains the effect of ligands and signaling receptors binding to NRP1 extracellular domain. Please see these changes in Lines 140-143.
Please check the text frequently references studies (e.g., Sema3A competing with VEGFR2 for VEGF binding, cleavage sites influencing activity). Are these studies cited appropriately with more recent or pivotal references?
Response: Thank you for your suggestion. These lines have been removed and updated with the latest work under the new Chapter 3. Please see the highlighted Chapter 3 at lines 201-252.
Some examples of semaphorins as tumor suppressors (e.g., Sema3B in lung cancer) are given, but others (like Sema3G) are briefly mentioned. Should their roles be elaborated for a more balanced discussion?
Response: Thank you for your suggestion. We updated most of the information about Sempahorin-NRP1 signaling in the new Chapter 3.
Elaborate on the physiological and pathological significance of VEGFA isoforms in angiogenesis and tumor biology.
Response: Thank you for your suggestion. This has been updated in the new Chapter 2.
Ensure consistent use of terms like semaphorins, Sema proteins, and NRP1 throughout the text.
Response: Thank you for your suggestion.

Reviewer 5 Report
Comments and Suggestions for Authors
The manuscript by Angom et al. entitled “Neuropilin-1: A Multifaceted Target for Cancer Therapy” represents a good-quality review providing in-depth analysis of physiological and pathological aspects of Neuropilin-1 functions with a focus on its involvement in cancer. The manuscript provides a robust analysis of available papers in the field and is well illustrated. However, some aspects of NRP1 are underrepresented. Provide more detailed data on the role of NRP1 in regulation of cell survival/apoptosis with signaling pathways affected, since this issue is critical in cancer, which is fundamentally considered a condition associated with dysregulated cell death. What is currently known about the contribution of NRP1 to non-apoptotic regulated cell death modalities such as necroptosis, pyroptosis, ferroptosis, etc.? NRP1 is involved in regulation of mitochondrial functions. This aspect is not covered in the manuscript.
Other major issues:
- Provide the list of abbreviations for Table 1. The same should be applied for Table 2.
- Subheadings 3: Anti-angiogenic functions of what?
Minor issues:
Line 42. Please remove the information that Figure 1 was created with Biorender and add it to the Figure legend.
Line 46. “migration” instead of “Migration”
Line 70. The full name for NRP1 was introduced above. Please use just the abbreviation
Lin 188. Which cell lines?
Author Response
Response to Reviewer 5:
We thank the reviewers for all the comments and valuable suggestions. The reviewers' comments have significantly improved our manuscript. In this revision, we have carefully addressed all the comments suggested by the reviewers. Please see our point-by-point response below.
Provide more detailed data on the role of NRP1 in regulation of cell survival/apoptosis with signaling pathways affected, since this issue is critical in cancer, which is fundamentally considered a condition associated with dysregulated cell death.
What is currently known about the contribution of NRP1 to non-apoptotic regulated cell death modalities such as necroptosis, pyroptosis, ferroptosis, etc.? NRP1 is involved in regulation of mitochondrial functions. This aspect is not covered in the manuscript.
Response: Thank you for the suggestion. Please refer to the new Chapter 6, titled “Other regulatory role of NRP1,” for detailed discussions about apoptosis and other types of cell death. Please refer to lines 382- 422.
Other major issues:
Provide the list of abbreviations for Table 1. The same should be applied to Table 2.
Response: Thank you for your suggestion. A glossary of complex terms is provided at the end of the article.
Subheadings 3: Anti-angiogenic functions of what?
Response: Thank you for this correction. This chapter has been revised and updated in the manuscript. Please see the new Chapter 5.
Minor issues:
Line 42. Please remove the information that Figure 1 was created with Bio render and add it to the Figure legend.
Response: This has been removed from the text and updated in the figure 1 legend. Please refer to line 103. Figure was created using Biorender. ( https://biorender.com.).
Line 46. “migration” instead of “Migration”
Response: This word has been changed in line 46.
Line 70. The full name for NRP1 was introduced above. Please use just the abbreviation
Response: Thank You. The manuscript has been revised.
Lin 188. Which cell lines?
Response: Thank you for this suggestion. We have included the information: small cell lung cancer cell line U2020.

Round 2
Reviewer 5 Report
Comments and Suggestions for Authors
The comments have been addressed.